# Online Training Through Time for Spiking Neural Networks

**Mingqing Xiao**[1], **Qingyan Meng**[2,3], **Zongpeng Zhang**[4], **Di He**[1], **Zhouchen Lin**[1,5,6][*]

[1]Key Lab. of Machine Perception (MoE), School of Intelligence Science and Technology,
Peking University
[2]The Chinese University of Hong Kong, Shenzhen
[3]Shenzhen Research Institute of Big Data
[4]Center for Data Science, Academy for Advanced Interdisciplinary Studies, Peking University
[5]Institute for Artificial Intelligence, Peking University
[6]Peng Cheng Laboratory, China
{mingqing_xiao, dihe, zlin}@pku.edu.cn, qingyanmeng@link.cuhk.edu.cn,
zongpeng.zhang98@gmail.com

## Abstract

Spiking neural networks (SNNs) are promising brain-inspired energy-efficient models. Recent progress in training methods has enabled successful deep SNNs on large-scale tasks with low latency. Particularly, backpropagation through time (BPTT) with surrogate gradients (SG) is popularly used to enable models to achieve high performance in a very small number of time steps. However, it is at the cost of large memory consumption for training, lack of theoretical clarity for optimization, and inconsistency with the online property of biological learning rules and rules on neuromorphic hardware. Other works connect the spike representations of SNNs with equivalent artificial neural network formulation and train SNNs by gradients from equivalent mappings to ensure descent directions. But they fail to achieve low latency and are also not online. In this work, we propose online training through time (OTTT) for SNNs, which is derived from BPTT to enable forward-in-time learning by tracking presynaptic activities and leveraging instantaneous loss and gradients. Meanwhile, we theoretically analyze and prove that the gradients of OTTT can provide a similar descent direction for optimization as gradients from equivalent mapping between spike representations under both feedforward and recurrent conditions. OTTT only requires constant training memory costs agnostic to time steps, avoiding the significant memory costs of BPTT for GPU training. Furthermore, the update rule of OTTT is in the form of three-factor Hebbian learning, which could pave a path for online on-chip learning. With OTTT, it is the first time that the two mainstream supervised SNN training methods, BPTT with SG and spike representation-based training, are connected, and meanwhile it is in a biologically plausible form. Experiments on CIFAR-10, CIFAR-100, ImageNet, and CIFAR10-DVS demonstrate the superior performance of our method on large-scale static and neuromorphic datasets in a small number of time steps. Our code is available at https://github.com/pkuxmq/OTTT-SNN.

## 1 Introduction

Spiking neural networks (SNNs) are regarded as the third generation of neural network models [1] and have gained increasing attention in recent years [2, 3, 4, 5, 6, 7, 8, 9, 10, 11, 12, 13]. SNNs are composed of brain-inspired spiking neurons that imitate biological neurons to transmit spikes

---

[*]Corresponding author.

36th Conference on Neural Information Processing Systems (NeurIPS 2022).

between each other. This allows event-based computation and enables efficient computation on neuromorphic hardware with low energy consumption [14, 15, 16].

However, the supervised training of SNNs is challenging due to the non-differentiable neuron model with discrete spike-generation procedures. Several kinds of methods are proposed to tackle the problem, and recent progress has empirically obtained successful results. Backpropagation through time (BPTT) with surrogate gradients (SG) is one of the mainstream methods which enables the training of deep SNNs with high performance on large-scale datasets (e.g., ImageNet) with extremely low latency (e.g., 4-6 time steps) [6, 10, 11, 13]. These approaches unfold the iterative expression of spiking neurons, backpropagate the errors through time [17], and use surrogate derivatives to approximate the gradient of the spiking function [3, 4, 18, 19, 20, 21, 22, 23]. As a result, during training, they suffer from significant memory costs proportional to the number of time steps, and the optimization with approximated surrogate gradients is not well guaranteed theoretically. Another branch of works builds the closed-form formulation for the spike representation of neurons, e.g. the (weighted) firing rate or spiking time, which is similar to conventional artificial neural networks (ANNs). Then SNNs can be either optimized by calculating the gradients from the equivalent mappings between spike representations [2, 24, 25, 26, 9, 27], or converted from a trained equivalent ANN counterpart [28, 29, 30, 31, 32, 7, 33, 8, 34]. The optimization of these methods is clearer than surrogate gradients. However, they require a larger number of time steps compared to BPTT with SG. Therefore, they suffer from high latency, and more energy consumption is required if the representation is rate-based. Another critical point for both methods is that they are indeed inconsistent with biological online learning, which is also the learning rule on neuromorphic hardware [15].

In this work, we develop a novel approach for training SNNs to achieve high performance with low latency, and maintain the online learning property to pave a path for learning on neuromorphic chips. We call our method online training through time (OTTT). We first derive OTTT from the commonly used BPTT with SG method by analyzing the temporal dependency and proposing to track the presynaptic activities in order to decouple this dependency. With the instantaneous loss calculation, OTTT can perform forward-in-time learning, i.e. calculations are done online in time without computing backward through the time. Then we theoretically analyze the gradients of OTTT and gradients of spike representation-based methods. We show that they have similar expressions and prove that they can provide the similar descent direction for the optimization problem formulated by spike representation. For the feedforward network condition, gradients are easily calculated and analyzed. For the recurrent network condition, we follow the framework in [12] that weighted firing rates will converge to an equilibrium state and gradients can be calculated by implicit differentiation. With this formulation, the gradients correspond to an approximation of gradients calculated by implicit differentiation, which can be proved to be a descent direction for the optimization problem as well [35, 36]. In this way, a connection between OTTT and spike representation-based methods is bridged. Finally, we show that OTTT is in the form of three-factor Hebbian learning rule [37], which could pave a path for online learning on neuromorphic chips. Our contributions include:

1. We propose online training through time (OTTT) for SNNs, which enables forward-in-time learning and only requires constant training memory agnostic to time steps, avoiding the large training memory costs of backpropagation through time (BPTT).

2. We theoretically analyze and connect the gradients of OTTT and gradients based on spike representations, and prove the descent guarantee for optimization under both feedforward and recurrent conditions.

3. We show that OTTT is in the form of three-factor Hebbian learning rule, which could pave a path for on-chip online learning. With OTTT, it is the first time that a connection between BPTT with SG, spike representation-based methods, and biological three-factor Hebbian learning is bridged.

4. We conduct extensive experiments on CIFAR-10, CIFAR-100, ImageNet, and CIFAR10-DVS, which demonstrate the superior results of our methods on large-scale static and neuromorphic datasets in a small number of time steps.

## 2  Related Work

**SNN Training Methods.**    As for supervised training of SNNs, there are two main research directions. One direction is to build a connection between spike representations (e.g. firing rates) of SNNs with equivalent ANN-like closed-form mappings. With the connection, SNNs can be converted from

ANNs [28, 29, 30, 31, 32, 7, 33, 8, 34, 38], or SNNs can be optimized by gradients calculated from equivalent mappings [2, 24, 25, 26, 9, 27]. Variants following this direction also include [12] which connects feedback SNNs with equilibrium states following fixed-point equations instead of closed-form mappings. These methods have a clearer descent direction for the optimization problem, but require a relatively large number of time steps, suffering from high latency and usually more energy consumption with rate based representation. Another direction is to directly calculate gradients based on the SNN computation. They follow the BPTT framework, and deal with the non-differentiable problem of spiking functions by applying surrogate gradients [3, 4, 18, 19, 20, 21, 6, 23, 10, 11, 13], or computing gradients with respect to spiking times based on the linear assumption [39, 40], or combining both [22]. BPTT with SG can achieve extremely low latency. However, it requires large training memory to maintain the computational graph unfolded along time, and it remains unknown why surrogate gradients work well. [10] empirically observed that surrogate gradients have a high similarity with numerical gradients, but it remains unclear theoretically. And gradients based on spiking times suffer from the "dead neuron" problem [3], so they should be combined with SG in practice [40, 22]. Meanwhile, methods in both directions are inconsistent with biological online learning, i.e. forward-in-time learning, to pave a path for learning on neuromorphic hardware. Differently, our proposed method avoids the above problems and maintain the online property.

**Online Training of Neural Networks.** In the research of recurrent neural networks (RNNs), there are several alternatives for BPTT to enable online learning. Particularly, real time recurrent learning (RTRL) [41] proposes to propagate partial derivatives of hidden states over parameters through time to enable forward-in-time calculation of gradients. Several recent works improve the memory costs of RTRL with approximation for more practical usage [42, 43, 44]. Another work proposes to online update parameters based on decoupled gradients with regularization at each time step [45]. However, these are all for RNNs and not tailored to SNNs. Several online training methods are proposed for SNNs [46, 47, 48], which are derived in the spirit of RTRL and simplified for SNNs. [49] leverages local losses and ignores temporal dependencies for online local training of SNNs, and [50] directly apply the method in [45] to train SNNs. However, these methods also leverage surrogate gradients without providing theoretical explanation for optimization. Meanwhile, [46, 49] use feedback alignment [51], [47] is limited to single-layer recurrent SNNs, and [48] requires much larger memory costs for eligibility traces, so they cannot scale to large-scale tasks. [50] requires a specially designed neuron model and more computation for parameter regularization, and also does not consider large tasks. Differently, our work explain the descent direction under both feedforward and recurrent conditions with convergent inputs, and is efficient and scalable to large-scale tasks including ImageNet classification.

## 3 Preliminaries

### 3.1 Spiking Neural Networks

Spiking neurons are brain-inspired models that transmit information by spike trains. Each neuron maintains a membrane potential $u$ and integrates input spike trains, which will generate a spike once $u$ exceeds a threshold. We consider the commonly used leaky integrate and fire (LIF) model, which describes the dynamics of the membrane potential as:

$$\tau_m \frac{du}{dt} = -(u - u_{rest}) + R \cdot I(t), \quad u < V_{th}, \tag{1}$$

where $I$ is the input current, $V_{th}$ is the threshold, and $R$ and $\tau_m$ are resistance and time constant, respectively. A spike is generated when $u$ reaches $V_{th}$ at time $t^f$, and $u$ is reset to the resting potential $u = u_{rest}$, which is usually set to be zero. The output spike train is defined using the Dirac delta function: $s(t) = \sum_{t^f} \delta(t - t^f)$.

A spiking neural network is composed of connected spiking neurons with connection coefficients. We consider a simple current model $I_i(t) = \sum_j w_{ij} s_j(t) + b_i$, where the subscript $i$ represents the $i$-th neuron, $w_{ij}$ is the weight from neuron $j$ to neuron $i$, and $b_i$ is a bias. The discrete computational form is:

$$\begin{cases} u_i[t+1] = \lambda(u_i[t] - V_{th} s_i[t]) + \sum_j w_{ij} s_j[t] + b_i, \\ s_i[t+1] = H(u_i[t+1] - V_{th}), \end{cases} \tag{2}$$

where $H(x)$ is the Heaviside step function, $s_i[t]$ is the spike train of neuron $i$ at discrete time step $t$, and $\lambda < 1$ is a leaky term (typically taken as $1 - \frac{1}{\tau_m}$). The constant $R$, $\tau_m$, and time step size are absorbed into the weights and bias. The reset operation is implemented by subtraction.

## 3.2 Previous SNN Training Methods

**Spike Representation.** The (weighted) firing rate or first spiking time of spiking neurons can be proved to follow ANN-like closed-form transformations [7, 26, 9, 12, 27]. We focus on the weighted firing rate [12, 27] which has connection with OTTT in this work. Define weighted firing rates and weighted average inputs $\mathbf{a}[t] = \frac{\sum_{\tau=1}^{t} \lambda^{t-\tau} \mathbf{s}[\tau]}{\sum_{\tau=1}^{t} \lambda^{t-\tau}}$, $\overline{\mathbf{x}}[t] = \frac{\sum_{\tau=0}^{t} \lambda^{t-\tau} \mathbf{x}[\tau]}{\sum_{\tau=0}^{t} \lambda^{t-\tau}}$ in the discrete condition. Given convergent weighted average inputs $\overline{\mathbf{x}}[t] \to \mathbf{x}^*$, it can be proved that $\mathbf{a}[t] \to \mathbf{a}^* = \sigma\left(\frac{1}{V_{th}} \mathbf{x}^*\right)$ with bounded random error, where $\sigma$ is a clamp function ($\sigma(x) = \min(\max(0, x), 1)$) in the discrete condition while a ReLU function in the continuous condition. For feedforward networks, the closed-form mapping between successive layers is established based on weighted firing rate after time $T$: $\mathbf{a}^{l+1}[T] \approx \sigma\left(\frac{1}{V_{th}}\left(\mathbf{W}^l \mathbf{a}^l[T] + \mathbf{b}^{l+1}\right)\right)$, and gradients are calculated with such spike representation: $\frac{\partial L}{\partial \mathbf{W}^l} = \frac{\partial L}{\partial \mathbf{a}^N[T]} \prod_{i=N-1}^{l+1} \frac{\partial \mathbf{a}^{i+1}[T]}{\partial \mathbf{a}^i[T]} \frac{\partial \mathbf{a}^{l+1}[T]}{\partial \mathbf{W}^l}$. For the recurrent condition, $\mathbf{a}[t]$ will converge to an equilibrium state following an implicit fixed-point equation, e.g. $\mathbf{a}^* = \sigma\left(\frac{1}{V_{th}}\left(\mathbf{W}\mathbf{a}^* + \mathbf{F}\mathbf{x}^* + \mathbf{b}\right)\right)$ for a single-layer network with input connections $F$ and contractive recurrent connections $W$, and gradients can be calculated based on implicit differentiation [12]. Let $\mathbf{a} = f_{\boldsymbol{\theta}}(\mathbf{a})$ denote the fixed-point equation ($\boldsymbol{\theta}$ are parameters). We have $\frac{\partial L}{\partial \boldsymbol{\theta}} = \frac{\partial L}{\partial \mathbf{a}[T]} \left(I - J_{f_{\boldsymbol{\theta}}}|_{\mathbf{a}[T]}\right)^{-1} \frac{\partial f_{\boldsymbol{\theta}}(\mathbf{a}[T])}{\partial \boldsymbol{\theta}}$, where $J_{f_{\boldsymbol{\theta}}}|_{\mathbf{a}[T]} = \frac{\partial f_{\boldsymbol{\theta}}(\mathbf{a}[T])}{\partial \mathbf{a}[T]}$ is the Jacobian of $f_{\boldsymbol{\theta}}$ at $\mathbf{a}[T]$.

**BPTT with SG.** BPTT unfolds the iterative update equation in Eq.(2) and backpropagates along the computational graph as shown in Fig. 1(a), 1(c). The gradients with $T$ time steps are calculated by [2]:

$$\frac{\partial L}{\partial \mathbf{W}^l} = \sum_{t=1}^{T} \frac{\partial L}{\partial \mathbf{s}^{l+1}[t]} \frac{\partial \mathbf{s}^{l+1}[t]}{\partial \mathbf{u}^{l+1}[t]} \left( \frac{\partial \mathbf{u}^{l+1}[t]}{\partial \mathbf{W}^l} + \sum_{\tau < t} \prod_{i=t-1}^{\tau} \left( \frac{\partial \mathbf{u}^{l+1}[i+1]}{\partial \mathbf{u}^{l+1}[i]} + \frac{\partial \mathbf{u}^{l+1}[i+1]}{\partial \mathbf{s}^{l+1}[i]} \frac{\partial \mathbf{s}^{l+1}[i]}{\partial \mathbf{u}^{l+1}[i]} \right) \frac{\partial \mathbf{u}^{l+1}[\tau]}{\partial \mathbf{W}^l} \right),$$

(3)

where $\mathbf{W}^l$ is the weight from layer $l$ to $l+1$ and $L$ is the loss. The non-differentiable terms $\frac{\partial \mathbf{s}^l[t]}{\partial \mathbf{u}^l[t]}$ will be replaced by surrogate derivatives, e.g. derivatives of rectangular or sigmoid functions [4]: $\frac{\partial s}{\partial u} = \frac{1}{a_1} \text{sign}\left(|u - V_{th}| < \frac{a_1}{2}\right)$ or $\frac{\partial s}{\partial u} = \frac{1}{a_2} \frac{e^{(V_{th}-u)/a_2}}{(1+e^{(V_{th}-u)/a_2})^2}$, where $a_1$ and $a_2$ are hyperparameters.

## 4 Online Training Through Time for SNNs

This section contains four sub-sections. In Section 4.1, we introduce our proposed OTTT by decoupling the temporal dependency of BPTT. Then in Section 4.2, we further connect the gradients of OTTT and spike representation-based methods, and prove that OTTT can provide a descent direction for optimization, which is not guaranteed by BPTT with SG. In Section 4.3, we discuss the relationship between OTTT and the three-factor Hebbian learning rule. Implementation details are presented in Section 4.4.

### 4.1 Derivation of Online Training Through Time

**Decouple temporal dependency.** As shown in Fig. 1(c), BPTT has to maintain the computational graph of previous time to backpropagate through time. We will decouple such temporal dependency to enable online gradient calculation, as illustrated in Fig. 1(d).

We first focus on the feedforward network condition. In this setting, all temporal dependencies lie in the dynamics of each spiking neuron, i.e. $\frac{\partial \mathbf{u}^{l+1}[i+1]}{\partial \mathbf{u}^{l+1}[i]}$ and $\frac{\partial \mathbf{u}^{l+1}[i+1]}{\partial \mathbf{s}^{l+1}[i]} \frac{\partial \mathbf{s}^{l+1}[i]}{\partial \mathbf{u}^{l+1}[i]}$ in Eq.(3). We consider the case that we do not apply surrogate derivatives to $\frac{\partial \mathbf{s}^{l+1}[i]}{\partial \mathbf{u}^{l+1}[i]}$ in such temporal dependency. Since the derivative of the Heaviside step function is 0 almost everywhere, we have $\frac{\partial \mathbf{u}^{l+1}[i+1]}{\partial \mathbf{s}^{l+1}[i]} \frac{\partial \mathbf{s}^{l+1}[i]}{\partial \mathbf{u}^{l+1}[i]} \approx 0$ [3].

---

[2]Note that we follow the numerator layout convention for derivatives, i.e. $\nabla_{\boldsymbol{\theta}} L = \left(\frac{\partial L}{\partial \boldsymbol{\theta}}\right)^\top$ is the gradient with the same dimension of $\boldsymbol{\theta}$.

[3]Note that this is consistent with some released implementations of BPTT with SG methods which detach the neuron reset operation from the computational graph and do not backpropagate gradients in this path [23, 11].

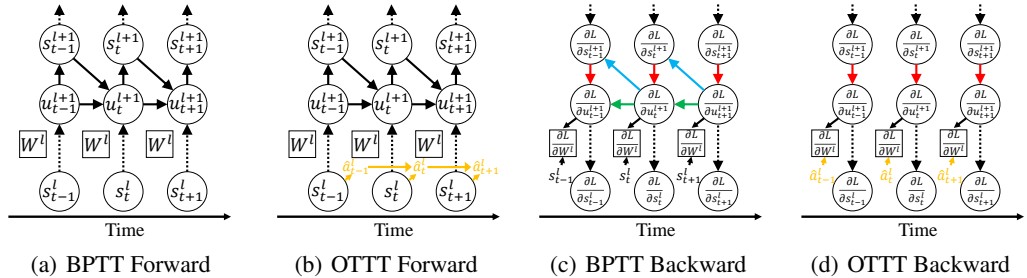

Figure 1: Illustration of the forward and backward procedures of BPTT and OTTT.

Then the dependency only includes $\frac{\partial \mathbf{u}^{l+1}[i+1]}{\partial \mathbf{u}^{l+1}[i]}$, which equals $\lambda \mathbf{I}$. Therefore, we have[2]:

$$\frac{\partial L}{\partial \mathbf{W}^l} = \sum_{t=1}^{T} \frac{\partial L}{\partial \mathbf{s}^{l+1}[t]} \frac{\partial \mathbf{s}^{l+1}[t]}{\partial \mathbf{u}^{l+1}[t]} \left( \sum_{\tau \leq t} \lambda^{t-\tau} \frac{\partial \mathbf{u}^{l+1}[\tau]}{\partial \mathbf{W}^l} \right), \nabla_{\mathbf{W}^l} L = \sum_{t=1}^{T} \mathbf{g}_{\mathbf{u}^{l+1}}[t] \left( \sum_{\tau \leq t} \lambda^{t-\tau} \mathbf{s}^l[\tau] \right)^{\top},$$

(4)

where $\mathbf{g}_{\mathbf{u}^{l+1}}[t] = \left( \frac{\partial L}{\partial \mathbf{s}^{l+1}[t]} \frac{\partial \mathbf{s}^{l+1}[t]}{\partial \mathbf{u}^{l+1}[t]} \right)^{\top}$ is the gradient for $\mathbf{u}^{l+1}[t]$. Based on Eq.(4), we can track presynaptic activities $\hat{\mathbf{a}}^l[t] = \sum_{\tau \leq t} \lambda^{t-\tau} \mathbf{s}^l[\tau]$ for each neuron during the forward procedure by $\hat{\mathbf{a}}^l[t+1] = \lambda \hat{\mathbf{a}}^l[t] + \mathbf{s}^l[t+1]$, so that when given $\mathbf{g}_{\mathbf{u}^{l+1}}[t]$, gradients at each time step can be calculated independently by $\nabla_{\mathbf{W}^l} L[t] = \mathbf{g}_{\mathbf{u}^{l+1}}[t] \hat{\mathbf{a}}^l[t]^{\top}$ without backpropagation through $\frac{\partial \mathbf{u}^{l+1}[i+1]}{\partial \mathbf{u}^{l+1}[i]}$.

As for the recurrent network condition, there are additional temporal dependencies due to the feedback connections between neurons. If there is feedback connection from layer $l_2$ to $l_1$ ($l_2 \geq l_1$), there would be terms such as $\frac{\partial \mathbf{u}^{l_1}[i+1]}{\partial \mathbf{s}^{l_2}[i]} \frac{\partial \mathbf{s}^{l_2}[i]}{\partial \mathbf{u}^{l_2}[i]} \frac{\partial \mathbf{u}^{l_2}[i]}{\partial \mathbf{u}^{l_1}[i]}$ in the calculation of gradients (note that Eq. (3) omit feedback connections for simplicity). We also consider not applying surrogate derivatives to $\frac{\partial \mathbf{s}^{l_2}[i]}{\partial \mathbf{u}^{l_2}[i]}$ in the temporal dependency so that gradients are not calculated in this path. Similar to the feedforward condition, we can derive that the gradients of the general weight $\mathbf{W}^{l_i \to l_j}$ from any layer $l_i$ to any layer $l_j$ can be calculated by $\nabla_{\mathbf{W}^{l_i \to l_j}} L[t] = \mathbf{g}_{\mathbf{u}^{l_j}}[t] \hat{\mathbf{a}}^{l_i}[t]^{\top}$ at each time step. A theoretical explanation for optimization will be presented in Section 4.2.

**Instantaneous Loss and Gradient.** Calculating online gradients, e.g. the above $\mathbf{g}_{\mathbf{u}^{l+1}}[t]$ for $\nabla_{\mathbf{W}^l} L[t]$, requires instantaneous computation of the loss at each time step. Previous typical loss for SNNs is based on the firing rate, e.g. $L_{fr} = \mathcal{L}\left(\frac{1}{T} \sum_{t=1}^{T} \mathbf{s}^N[t], \mathbf{y}\right)$, where $\mathbf{y}$ is the label, $\mathbf{s}^N[t]$ is the spike at the last layer, and $\mathcal{L}$ can take cross-entropy loss. This loss depends on all time steps and does not support online gradients. We leverage the instantaneous loss and calculate the above $\mathbf{g}_{\mathbf{u}^{l+1}}[t]$ as:

$$L[t] = \frac{1}{T}\mathcal{L}\left(\mathbf{s}^N[t], \mathbf{y}\right), \quad \mathbf{g}_{\mathbf{u}^{l+1}}[t] = \left( \frac{\partial L[t]}{\partial \mathbf{s}^N[t]} \prod_{i=N-1}^{l+1} \frac{\partial \mathbf{s}^{i+1}[t]}{\partial \mathbf{s}^i[t]} \frac{\partial \mathbf{s}^{l+1}[t]}{\partial \mathbf{u}^{l+1}[t]} \right)^{\top}.$$

(5)

The total loss $L := \sum_{t=1}^{T} L[t]$ is the upper bound of $L_{fr}$ when $\mathcal{L}$ is a convex function such as cross-entropy. Then gradients can be calculated independently at each time step, as shown in Fig. 1(d). We apply surrogate derivatives for $\frac{\partial \mathbf{s}^l[t]}{\partial \mathbf{u}^l[t]}$ in this calculation, which will be explained in Section 4.2.

Since gradients are calculated instantaneously at each time step, OTTT does not require maintaining the unfolded computational graph and only requires constant training memory costs agnostic to time steps. Note that instantaneous gradients of OTTT will be different from BPTT with the instantaneous loss for multi-layer or recurrent networks, as we do not consider future influence in the instantaneous calculation: BPTT considers terms such as $\frac{\partial L[t']}{\partial \mathbf{u}^N[t']} \frac{\partial \mathbf{u}^N[t']}{\partial \mathbf{u}^N[t]} \prod_{i=N-1}^{l} \frac{\partial \mathbf{u}^{i+1}[t]}{\partial \mathbf{u}^i[t]}$ ($t' > t$) for $\mathbf{u}^l[t]$ while we do not. The equivalence of OTTT and BPTT only holds for the last layer, and we do not seek the exact equivalence to BPTT with SG which is theoretically unclear, but will build the connection with spike representations and prove the descent guarantee. Also, note that the tracked presynaptic activities are similar to the biologically plausible "eligibility traces" in the literature [46, 47, 52], and we will provide a more solid theoretical grounding for optimization in Section 4.2.

## 4.2 Connection with Spike Representation-Based Methods for Descent Directions

In this section, we connect gradients of OTTT and spike representation-based methods, and prove that OTTT can provide a descent direction for optimization under feedforward and recurrent conditions with convergent inputs.

**Feedforward Networks.** As introduced in Section 3.2, with convergent inputs, methods based on spike representations establish closed-form mappings between successive layers with weighted firing rate $\mathbf{a}[t] = \frac{\sum_{\tau=1}^{t} \lambda^{t-\tau} \mathbf{s}[\tau]}{\sum_{\tau=1}^{t} \lambda^{t-\tau}}$ as $\mathbf{a}^{l+1}[T] \approx \sigma\left(\frac{1}{V_{th}}\left(\mathbf{W}^l \mathbf{a}^l[T] + \mathbf{b}^{l+1}\right)\right)$, and calculate gradients by $\frac{\partial L}{\partial \mathbf{W}^l} = \frac{\partial L}{\partial \mathbf{a}^N[T]} \prod_{i=N-1}^{l+1} \frac{\partial \mathbf{a}^{i+1}[T]}{\partial \mathbf{a}^i[T]} \frac{\partial \mathbf{a}^{l+1}[T]}{\partial \mathbf{W}^l}$. Note that $\mathbf{a}[t]$ is similar to the tracked presynaptic activities $\hat{\mathbf{a}}[t] = \sum_{\tau=1}^{t} \lambda^{t-\tau} \mathbf{s}[\tau]$ in OTTT. We can obtain:

$$(\nabla_{\mathbf{W}^l} L_{sr})_{sr} = \sum_{t=1}^{T}\left(\left(\frac{1}{T}\frac{1}{\lambda^{T-t}}\frac{\partial L_{sr}}{\partial \mathbf{s}^N[t]}\prod_{i=N-1}^{l+1}\frac{\partial \mathbf{a}^{i+1}[T]}{\partial \mathbf{a}^i[T]}\right)^{\top}\odot \mathbf{d}^{l+1}[T]\right)\hat{\mathbf{a}}^l[T]^{\top}, \quad (6)$$

where $L_{sr}$ is the loss based on spike representation, $\mathbf{d}^{l+1}[T] = \sigma'\left(\frac{1}{V_{th}}\left(\mathbf{W}^l \mathbf{a}^l[T] + \mathbf{b}^{l+1}\right)\right)$, and '$\odot$' is element-wise product. The detailed derivation can be found in Appendix A.

It can be easily seen that Eq. (6) has a similar form as gradients in Eq. (4), (5), and we build the connection between them in three steps. For sake of clarify, in the following, we denote gradients of OTTT and spike representation by $\nabla_{\mathbf{W}^l} L$ and $(\nabla_{\mathbf{W}^l} L_{sr})_{sr}$, respectively.

In the first step, we can use appropriate surrogate derivatives of $\frac{\partial \mathbf{s}_i^{l+1}[t]}{\partial \mathbf{u}_i^{l+1}[t]} (t = 1, \cdots, T)$ to approximate $\mathbf{d}_i^{l+1}[T]$. As introduced in Section 3.2, $\sigma$ is a clamp function ($\sigma(x) = \min(\max(0, x), 1)$) in the discrete condition while a ReLU function in the continuous condition. Then $\mathbf{d}^{l+1}[T]$ almost equals $\text{sign}\left(|\mathbf{W}^l \mathbf{a}^l[T] + \mathbf{b}^{l+1} - V_{th}| < V_{th}\right)$ if we slightly relax the clamp bound caused by the discretization, and this can be approximated by $\text{diag}\left(\frac{\partial \mathbf{s}^{l+1}[t]}{\partial \mathbf{u}^{l+1}[t]}\right) = \text{sign}\left(|\mathbf{u}^{l+1}[t] - V_{th}| < V_{th}\right)$ at each time step. Note that surrogate derivatives here approximate the well-defined derivative of the mapping function between $\mathbf{a}[T]$, rather than the pseudo derivative of the non-differentiable Heaviside step function as in BPTT with SG. The approximation is exact except the rare case that averagely a neuron generate spikes (i.e. the average input is positive) while sometimes the membrane potential is less than $u_{rest}$ (or the reverse). For simplicity in the following theoritical analysis, we assume the equivalence between surrogate derivatives and $\mathbf{d}[T]$.

**Assumption 1.** $\forall l = 1, \cdots, N, t = 1, \cdots, T, \text{diag}\left(\frac{\partial \mathbf{s}^{l+1}[t]}{\partial \mathbf{u}^{l+1}[t]}\right) = \mathbf{d}^{l+1}[T]$.

In the second step, we take $L_{sr}$ in Eq. (6) as $L_{srup} = \frac{1}{\sum_{\tau=0}^{T-1}\lambda^\tau}\sum_{t=1}^{T}\lambda^{T-t}\mathcal{L}(\mathbf{s}^N[t], \mathbf{y})$ so that $\frac{1}{T}\frac{1}{\lambda^{T-t}}\frac{\partial L_{sr}}{\partial \mathbf{s}^N[t]}$ in Eq. (6) aligns with $\frac{\partial L[t]}{\partial \mathbf{s}^N[t]}$ in Eq. (5) except a constant term. Note that $L_{srup}$ is an upper bound of the common loss $L'_{sr} = \mathcal{L}(\mathbf{a}^N[T], \mathbf{y}) = \mathcal{L}\left(\frac{\sum_{t=1}^{T}\lambda^{T-t}\mathbf{s}^N[t]}{\sum_{t=1}^{t}\lambda^{T-t}}, \mathbf{y}\right)$ if $\mathcal{L}$ is a convex function. Then let $\hat{\mathbf{g}}_{\mathbf{u}^{l+1}}[t] = \left(\frac{\partial \mathcal{L}(\mathbf{s}^N[t], \mathbf{y})}{\partial \mathbf{s}^N[t]}\prod_{i=N-1}^{l+1}\frac{\partial \mathbf{s}^{i+1}[t]}{\partial \mathbf{s}^i[t]}\frac{\partial \mathbf{s}^{l+1}[t]}{\partial \mathbf{u}^{l+1}[t]}\right)^{\top}$, with Assumption 1 we have: $\nabla_{\mathbf{W}^l} L = \frac{1}{T}\sum_{t=1}^{T}\hat{\mathbf{g}}_{\mathbf{u}^{l+1}}[t]\hat{\mathbf{a}}^l[t]^{\top}$ and $(\nabla_{\mathbf{W}^l} L_{sr})_{sr} = \frac{1}{T}\frac{1}{\sum_{\tau=0}^{T-1}\lambda^\tau}\sum_{t=1}^{T}\hat{\mathbf{g}}_{\mathbf{u}^{l+1}}[t]\hat{\mathbf{a}}^l[T]^{\top}$. Please refer to Appendix A for details.

In the third step, we handle the remaining difference that $\nabla_{\mathbf{W}^l} L$ leverages instantaneous presynaptic activities $\hat{\mathbf{a}}[t]$ during calculation while $(\nabla_{\mathbf{W}^l} L_{sr})_{sr}$ uses the final $\hat{\mathbf{a}}^l[T]$ after time $T$. Note that the weighted firing rate gradually converges $\mathbf{a}[t] \to \mathbf{a}^*$ with bounded random error. Suppose the errors $\boldsymbol{\epsilon}^l[t] = \mathbf{a}^l[t] - \mathbf{a}^l[T]$ are small ($l = 0$ represents inputs, i.e. $\mathbf{a}^0[t] = \overline{\mathbf{x}}[t]$), then we have that $-\nabla_{\mathbf{W}^l} L$ can provide a descent direction, as shown in Theorem 1.

**Theorem 1.** *If Assumption 1 holds, $V_{th} = 1$, and the errors $\boldsymbol{\epsilon}^l[t] = \mathbf{a}^l[t] - \mathbf{a}^l[T]$ are small such that $\left\|\sum_{t=1}^{T}\hat{\mathbf{g}}_{\mathbf{u}^{l+1}}[t]\boldsymbol{\epsilon}^l[t]^{\top}\right\| < \left\|\sum_{t=1}^{T}\hat{\mathbf{g}}_{\mathbf{u}^{l+1}}[t]\mathbf{a}^l[T]^{\top}\right\| - \left\|\sum_{t=1}^{T}\frac{\lambda^t(1-\lambda^{T-t})}{1-\lambda^T}\hat{\mathbf{g}}_{\mathbf{u}^{l+1}}[t]\mathbf{a}^l[t]^{\top}\right\|$ when $(\nabla_{\mathbf{W}^l} L_{sr})_{sr} \neq \mathbf{0}$, then we have $\langle \nabla_{\mathbf{W}^l} L, (\nabla_{\mathbf{W}^l} L_{sr})_{sr}\rangle > 0$.*

For the proof and discussion of the assumption please refer to Appendix A. With this conclusion, we can explain the descent direction of gradient descent by OTTT for the optimization problem

formulated by spike representation. Some random error can be viewed as randomness for stochastic optimization.

**Recurrent Networks.** For networks with feedback connections, we first consider the single-layer condition for simplicity (see Appendix A for general conditions). We consider feedforward connections $\mathbf{F}$ from inputs to neurons and contractive recurrent connections $\mathbf{W}$ between neurons. As introduced in Section 3.2, given convergent inputs $\overline{\mathbf{x}}[t] \to \mathbf{x}^*$, $\mathbf{a}[t]$ of neurons will converge to an equilibrium state $\mathbf{a}^* = f_{\boldsymbol{\theta}}(\mathbf{a}^*) = \sigma\left(\frac{1}{V_{th}}\left(\mathbf{W}\mathbf{a}^* + \mathbf{F}\mathbf{x}^* + \mathbf{b}\right)\right)$ with bounded random error, and gradients are calculated as $(\nabla_{\boldsymbol{\theta}} L_{sr})_{sr} = \left(\frac{\partial L_{sr}}{\partial \boldsymbol{\theta}}\right)^{\top} = \left(\frac{\partial L_{sr}}{\partial \mathbf{a}[T]}\left(I - J_{f_{\boldsymbol{\theta}}}|_{\mathbf{a}[T]}\right)^{-1}\frac{\partial f_{\boldsymbol{\theta}}(\mathbf{a}[T])}{\partial \boldsymbol{\theta}}\right)^{\top}$, where $\boldsymbol{\theta} \in \{\mathbf{W}, \mathbf{F}, \mathbf{b}\}$. We consider replacing the inverse Jacobian by an identity matrix: $(\widetilde{\nabla_{\boldsymbol{\theta}} L_{sr}})_{sr} = \left(\frac{\partial L_{sr}}{\partial \mathbf{a}[T]}\frac{\partial f_{\boldsymbol{\theta}}(\mathbf{a}[T])}{\partial \boldsymbol{\theta}}\right)^{\top}$. Previous works have proved that this gradient can provide a descent direction for the optimization problem [35, 36], i.e. $\left\langle (\widetilde{\nabla_{\boldsymbol{\theta}} L_{sr}})_{sr}, (\nabla_{\boldsymbol{\theta}} L_{sr})_{sr} \right\rangle > 0$. It has a similar form as the OTTT gradient: $\nabla_{\mathbf{W}} L = \sum_{t=1}^{T} \mathbf{g}_{\mathbf{u}}[t]\hat{\mathbf{a}}[t]^{\top}$ and $(\widetilde{\nabla_{\mathbf{W}} L_{sr}})_{sr} = \sum_{t=1}^{T}\left(\frac{1}{T}\frac{1}{\lambda^{T-t}}\frac{\partial L_{sr}}{\partial \mathbf{s}[t]}^{\top} \odot \mathbf{d}[T]\right)\hat{\mathbf{a}}[T]^{\top}$. Similarly, we can prove the descent guarantee for OTTT as shown in Theorem 2 . For details refer to Appendix A.

**Theorem 2.** *If Assumption 1 holds, $V_{th} = 1$, $\left\|J_{f_{\boldsymbol{\theta}}}|_{\mathbf{a}[T]}\right\| \le \eta < \frac{\sigma_{min}^2}{\sigma_{max}^2}$, where $\sigma_{max}$ and $\sigma_{min}$ are the maximal and minimal singular value of $\frac{\partial f_{\boldsymbol{\theta}}}{\partial \boldsymbol{\theta}}|_{\mathbf{a}[T]}$, and the errors $\boldsymbol{\epsilon}^1[t] = \mathbf{a}[t] - \mathbf{a}[T], \boldsymbol{\epsilon}^0[t] = \overline{\mathbf{x}}[t] - \overline{\mathbf{x}}[T]$ are small such that $\left\|\sum_{t=1}^{T} \hat{\mathbf{g}}_{\mathbf{u}}[t]\boldsymbol{\epsilon}^l[t]^{\top}\right\| < \frac{\sigma_{min}^2 - \eta\sigma_{max}^2}{\sigma_{max}}\left\|\sum_{t=1}^{T}\frac{\partial \mathcal{L}(\mathbf{s}[t], \mathbf{y})}{\partial \mathbf{s}[t]}\left(I - J_{f_{\boldsymbol{\theta}}}|_{\mathbf{a}[T]}\right)^{-1}\right\| - \left\|\sum_{t=1}^{T}\frac{\lambda^t(1-\lambda^{T-t})}{1-\lambda^T}\hat{\mathbf{g}}_{\mathbf{u}}[t]\mathbf{a}^l[t]^{\top}\right\|$ (where $l = 0, 1$, $\mathbf{a}^1[t]$ and $\mathbf{a}^0[t]$ represent $\mathbf{a}[t]$ and $\overline{\mathbf{x}}[t]$, respectively) when $(\nabla_{\boldsymbol{\theta}} L_{sr})_{sr} \ne \mathbf{0}$, then we have $\langle \nabla_{\boldsymbol{\theta}} L, (\nabla_{\boldsymbol{\theta}} L_{sr})_{sr}\rangle > 0$, where $\boldsymbol{\theta}$ are parameters in the network.*

### 4.3 Connection with Three-factor Hebbian Learning Rule

By explicitly writing the instantaneous gradients of OTTT for the general weight from layer $l_i$ to $l_j$, $\nabla_{\mathbf{W}^{l_i \to l_j}} L[t] = \mathbf{g}_{\mathbf{u}^{l_j}}[t]\hat{\mathbf{a}}^{l_i}[t]^{\top}$, and dive into connections between any two neurons $i$ and $j$, we have:

$$\nabla_{W_{i,j}} L[t] = \hat{a}_i[t] f(u_j[t])\delta_j[t], \tag{7}$$

where $\hat{a}_i[t]$ is the tracked presynaptic activity, $f(u_j[t])$ is the surrogate derivative function which can represent the change rate of the postsynaptic activity as analyzed in Section 4.2, and $\delta_j[t] = g_{s_j}[t]$ is the gradient for neuron output $s_j[t]$ which represents a global modulator. This is a kind of three-factor Hebbian learning rule [37] and the weight can be updated locally with a global signal. The error signal $\delta_j[t]$ can be propagated in an error feedback path simultaneously with feedforward propagation, which is shown biologically plausible with high-frequency bursts [53]. Note that the analysis in Section 4.2 still holds if we consider the delay of the propagation of the error signal, i.e. the update is based on $\hat{a}_i[t + \Delta t]f(u_j[t + \Delta t])\delta_j[t]$.

### 4.4 Implementation Details

As introduced in Section 4.1, we will calculate instantaneous gradients $\nabla_{\mathbf{W}^{l_i \to l_j}} L[t] = \mathbf{g}_{\mathbf{u}^{l_j}}[t]\hat{\mathbf{a}}^{l_i}[t]^{\top}$ at each time step. We can choose to immediately update parameters before the calculation of the next time step, which we denote as OTTT$_O$, or we can accumulate the gradients by $T$ time steps and then update parameters, which we denote as OTTT$_A$. For OTTT$_O$, we assume that the online update is small and has negligible affects for the following calculation. Pseudo-codes are in Appendix B.

An important issue in practice is that previous BPTT with SG works leverage batch normalization (BN) along the temporal dimension to achieve high performance with extremely low latency on large-scale datasets [6, 10, 11, 13], which requires calculating the mean and variance statistics for all time steps during the forward procedure. This technique intrinsically prevents online gradients and has to suffer from large memory costs. To overcome this shortcoming, we do not use BN, but borrow the idea from normalization-free ResNets (NF-ResNets) [54, 55] to replace batch normalization by scaled weight standardization (sWS) [56]. sWS standardizes weights by $\hat{\mathbf{W}}_{i,j} = \gamma \cdot \frac{\mathbf{W}_{i,j} - \mu_{\mathbf{W}_{i,\cdot}}}{\sigma_{\mathbf{W}_{i,\cdot}}\sqrt{N}}$, and the scale $\gamma$ is determined by analyzing the signal propagation with different activation functions. We apply sWS for VGG [57] and NF-ResNet architectures in our experiments. For details please refer to Appendix C.

# 5 Experiments

In this section, we conduct extensive experiments on CIFAR-10 [58], CIFAR100 [58], ImageNet [59], CIFAR10-DVS [60], and DVS128-Gesture [61] to demonstrate the superior performance of our proposed method on large-scale static and neuromorphic datasets. We leverage the VGG network architecture (64C3-128C3-AP2-256C3-256C3-AP2-512C3-512C3-AP2-512C3-512C3-GAP-FC) for experiments on CIFAR-10, CIFAR-100, CIFAR10-DVS, and DVS128-Gesture, and the NF-ResNet-34 [54] network architecture for experiments on ImageNet. For all our SNN models, we set $V_{th} = 1$ and $\lambda = 0.5$. Please refer to Appendix C for training details.

## 5.1 Comparison of Training Memory Costs

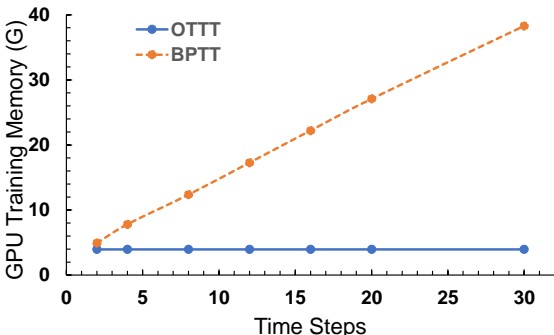

Figure 2: Comparison of training memory costs between OTTT and BPTT under different time steps.

A major advantage of OTTT over BPTT is that OTTT does not require backpropagation along the temporal dimension and therefore only requires constant training memory costs agnostic to time steps, which avoids the large memory costs of BPTT. We verify this by training the VGG network on CIFAR-10 with batch size 128 under different time steps and calculating the memory costs on the GPU. As shown in Fig. 2, the training memory of BPTT grows linearly with time steps, while OTTT maintains the constant memory (both $\text{OTTT}_A$ and $\text{OTTT}_O$). Even with a small number of time steps, e.g. 6, OTTT can reduce the memory costs by $2 \sim 3\times$. This advantage may also allow training acceleration of SNNs by larger batch sizes with the same computational resources.

## 5.2 Comparison of Performance

We conduct experiments on both large-scale static and neuromorphic datasets. We first verify the effectiveness of the surrogate derivative $\frac{\partial \mathbf{s}^{l+1}[t]}{\partial \mathbf{u}^{l+1}[t]} = \text{sign}\left(|\mathbf{u}^{l+1}[t] - V_{th}| < V_{th}\right)$ in Section 4.2. For the VGG (sWS) network on CIFAR-10, $\text{OTTT}_A$ and $\text{OTTT}_O$ achieves 91.46% and 92.47% test accuracy respectively. We empirically observe that applying the sigmoid-like surrogate derivative achieves a higher generalization performance, e.g. $\text{OTTT}_A$ and $\text{OTTT}_O$ achieves 93.58% and 93.73% test accuracy respectively under the same random seed, and a possible reason may be that this introduces some noise for the approximation to regularize the training and improve the generalization. Therefore, in the following performance evaluation, we take sigmoid-like surrogate derivative for OTTT. As shown in Table 1, both $\text{OTTT}_A$ and $\text{OTTT}_O$ achieve satisfactory performance on all datasets, and compared with BPTT under the same training settings, OTTT achieves higher performance. The proposed OTTT also achieves promising performance on all datasets compared with other representative conversion and direct training methods. Besides, it shows that the performance gap between our SNN model and ANN is around 0.7% and 2.08% on CIFAR-10 and CIFAR-100, respectively. Usually, SNNs with a very small number of time steps do not reach the performance of equivalent ANNs due to the information propagation with discrete spikes rather than floating-point numbers. The results of our model with 6 time steps are competitive.

We also evaluate our method on the DVS128-Gesture dataset, which is more time-varying with different hand gestures recorded by a DVS camera. As shown in Table 2, our method can achieve the same high performance as BPTT. While our theoretical analysis mainly focus on convergent inputs (e.g. static images or neuromorphic inputs converted from images like CIFAR10-DVS), the results show that our method can also work well for time-varying inputs.

Table 1: Performance on CIFAR-10, CIFAR-100, ImageNet, and CIFAR10-DVS. Results are based on 3 runs of experiments (except ImageNet). Our OTTT is mainly compared with BPTT under the same settings, and is also compared with other representative conversion and direct training methods.

| Dataset | Method | Network structure | Params | Time steps | Mean±Std (Best) |
|---|---|---|---|---|---|
| CIFAR-10 | ANN-SNN [7] | VGG-16 | 40M | 16 | (92.29%) |
| | BPTT [6] | ResNet-19 (tdBN) | 14.5M | 6 | (93.16%) |
| | BPTT [23] | 9-layer CNN (PLIF, BN) | 36M | 8 | (93.50%) |
| | BPTT | VGG (sWS) | 9.2M | 6 | 92.78±0.34 (93.23%) |
| | **OTTT$_A$ (ours)** | VGG (sWS) | 9.2M | 6 | **93.52±0.06% (93.58%)** |
| | **OTTT$_O$ (ours)** | VGG (sWS) | 9.2M | 6 | **93.49±0.17% (93.73%)** |
| | ANN | VGG (sWS) | 9.2M | N.A. | (94.43%) |
| CIFAR-100 | ANN-SNN [7] | VGG-16 | 40M | 400-600 | (70.55%) |
| | Hybrid Training [31] | VGG-11 | 36M | 125 | (67.87%) |
| | DIET-SNN [62] | VGG-16 | 40M | 5 | (69.67%) |
| | BPTT | VGG (sWS) | 9.3M | 6 | 69.06±0.07 (69.15%) |
| | **OTTT$_A$ (ours)** | VGG (sWS) | 9.3M | 6 | **71.05±0.04% (71.11%)** |
| | **OTTT$_O$ (ours)** | VGG (sWS) | 9.3M | 6 | **71.05±0.06% (71.11%)** |
| | ANN | VGG (sWS) | 9.3M | N.A. | (73.19%) |
| ImageNet | ANN-SNN [8] | ResNet-34 | 22M | 32 | (64.54%) |
| | Hybrid Training [31] | ResNet-34 | 22M | 250 | (61.48%) |
| | BPTT [6] | ResNet-34 (tdBN) | 22M | 6 | (63.72%) |
| | **OTTT$_A$ (ours)** | NF-ResNet-34 | 22M | 6 | **(65.15%)** |
| | **OTTT$_O$ (ours)** | NF-ResNet-34 | 22M | 6 | **(64.16%)** |
| DVS-CIFAR10 | Tandem Learning [9] | CifarNet | 45M | 20 | (65.59%) |
| | BPTT [6] | ResNet-19 (tdBN) | 14.5M | 10 | (67.80%) |
| | BPTT [23] | 7-layer CNN (PLIF, BN) | 1.1M | 20 | (74.80%) |
| | BPTT | VGG (sWS) | 9.2M | 10 | 72.60±1.26 (73.90%) |
| | **OTTT$_A$ (ours)** | VGG (sWS) | 9.2M | 10 | **76.27±0.05% (76.30%)** |
| | **OTTT$_O$ (ours)** | VGG (sWS) | 9.2M | 10 | **76.63±0.34% (77.10%)** |

Table 2: Performance on DVS128-Gesture.

| Method | Network structure | Time steps | Accuracy |
|---|---|---|---|
| SLAYER [3] | 8-layer CNN | 300 | 93.64±0.49% |
| DECOLLE [49] | 3-layer CNN | 1800 | 95.54±0.16% |
| BPTT [23] | 8-layer CNN (PLIF, BN) | 20 | 97.57% |
| BPTT [23] | 8-layer CNN (LIF, BN) | 20 | 96.88% |
| BPTT | VGG (sWS) | 20 | 96.88% |
| **OTTT$_A$ (ours)** | VGG (sWS) | 20 | 96.88% |

## 5.3 Effectiveness for Recurrence

As introduced in Section 4, the proposed OTTT is also valid for networks with feedback connections. Previous works have shown that adding feedback connections can improve the performance of SNNs without much additional costs, especially on the CIFAR-100 datasets [12, 63]. Therefore, we conduct experiments on CIFAR-100 with the VGG-F network architecture which simply adds a feedback connection from the last feature layer to the first feature layer following [12], and this weight is zero-intialized. As shown in Table 3, the training of VGG-F is valid and VGG-F achieves a higher performance than VGG due to the introduction of feedback connections. Results in Appendix D show that the improvement of OTTT from feedback connections is more significant than that of BPTT. The architectures with feedback connections can be further improved with neural architecture search [63].

## 5.4 Effectiveness for Training with Batch Size 1

To further study the online training, i.e. not only online in time but also one sample per training, which is consistent with biological learning and learning on neuromorphic hardware, we verify the effectiveness for training with batch size 1. The VGG network on CIFAR-10 is studied, and batch size 1 is compared with the default batch size 128 under the same random seed. Models are only trained for 20 epochs due to the relatively long training time with batch size 1. As shown in Table 4, training with one sample per iteration is still valid, indicating the potential to conduct full online training with the proposed OTTT.

Table 3: Performance on CIFAR-100 for VGG and VGG-F trained by $OTTT_O$. Results are based on 3 runs of experiments.

| Network structure | Params | Mean±Std (Best) |
|---|---|---|
| VGG | 9.3M | 71.05±0.06% (71.11%) |
| VGG-F | 9.6M | 72.63±0.23% (72.94%) |

Table 4: Performance of VGG on CIFAR-10 with different batch sizes for 20 epochs under the same random seed.

| Method | Batch Size | Accuracy |
|---|---|---|
| $OTTT_A$ / $OTTT_O$ | 128 | 88.20% / 88.62% |
| $OTTT_A$ / $OTTT_O$ | 1 | 88.07% / 88.50% |

### 5.5 Influence of Inference Time Steps

We study the influence of inference time steps on ImageNet as shown in Fig. 3. It illustrates that the model trained with time step 6 can achieve higher performance with more inference time steps.

### 5.6 Firing Rate Statistics

We study the firing rate statistics of the models trained by OTTT and BPTT, as shown in Fig. 4. It demonstrates that models trained by OTTT have higher firing rates in first layers while lower firing rates in later layers compared with BPTT. Overall the firing rate is around 0.19 and with 6 time steps each neuron averagely generate 1.1 spikes, indicating the low energy consumption. Considering that each neuron has more synaptic operations in later layers than first layers (because the channel size is increasing with layers), the synaptic operations of models trained by OTTT and BPTT are about the same ($1.98 \times 10^8$ vs $1.93 \times 10^8$). More results please refer to Appendix D.

Figure 3: Influence of inference time steps for the model trained with 6 time steps on ImageNet.

Figure 4: The average firing rates for the models trained by OTTT and BPTT on CIFAR-10.

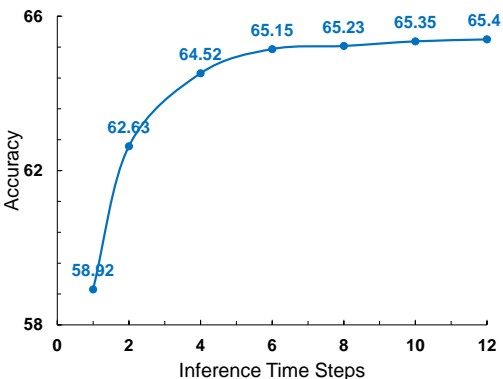
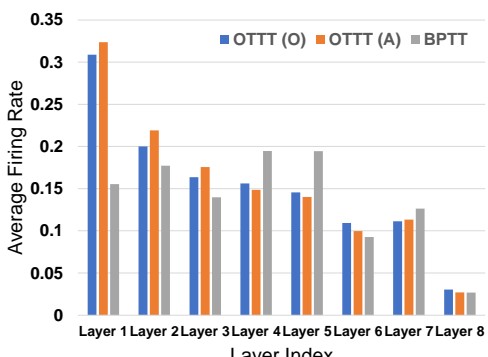

## 6 Conclusion

In this work, we propose a new training method, online training through time (OTTT), for spiking neural networks. We first derive OTTT from BPTT with SG by decoupling the temporal dependency with the tracked pre-synaptic activities, which only requires constant training memory agnostic to time steps and avoids the large training memory costs of BPTT. Then we theoretically analyze and connect the gradients of OTTT and gradients of methods based on spike representations, and prove the descent guarantee of OTTT for the optimization problem under both feedforward and recurrent network conditions. Additionally, we show that OTTT is in the form of three-factor Hebbian learning rule, which is the first to connect BPTT with SG, spike representation-based methods, and biological learning rules. Extensive experiments demonstrate the superior performance of our methods on large-scale static and neuromorphic datasets in a small number of time steps.

## Acknowledgement

Z. Lin was supported by the major key project of PCL (No. PCL2021A12), the NSF China (No.s 62276004 and 61731018), and Project 2020BD006 supported by PKU-Baidu Fund.

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
