# Supplementary Materials for: Online Training Through Time for Spiking Neural Networks

**Mingqing Xiao**[1], **Qingyan Meng**[2,3], **Zongpeng Zhang**[4], **Di He**[1], **Zhouchen Lin**[1,5,6][*]

[1]Key Lab. of Machine Perception (MoE), School of Intelligence Science and Technology,
Peking University
[2]The Chinese University of Hong Kong, Shenzhen
[3]Shenzhen Research Institute of Big Data
[4]Center for Data Science, Academy for Advanced Interdisciplinary Studies, Peking University
[5]Institute for Artificial Intelligence, Peking University
[6]Peng Cheng Laboratory, China
{mingqing_xiao, dihe, zlin}@pku.edu.cn, qingyanmeng@link.cuhk.edu.cn,
zongpeng.zhang98@gmail.com

## A Detailed Derivation and Proofs

### A.1 Derivation of Eq. (6)

Since $\mathbf{a}[t] = \frac{\sum_{\tau=1}^{t} \lambda^{t-\tau} \mathbf{s}[\tau]}{\sum_{\tau=1}^{t} \lambda^{t-\tau}}, \hat{\mathbf{a}}[t] = \sum_{\tau=1}^{t} \lambda^{t-\tau} \mathbf{s}[\tau], \mathbf{a}^{l+1}[T] \approx \sigma\left(\frac{1}{V_{th}}\left(\mathbf{W}^l \mathbf{a}^l[T] + \mathbf{b}^{l+1}\right)\right)$, $\mathbf{d}^{l+1}[T] = \sigma'\left(\frac{1}{V_{th}}\left(\mathbf{W}^l \mathbf{a}^l[T] + \mathbf{b}^{l+1}\right)\right)$, $\left(\frac{\partial L_{sr}}{\partial \mathbf{W}^l}\right)_{sr} = \frac{\partial L_{sr}}{\partial \mathbf{a}^N[T]} \prod_{i=N-1}^{l+1} \frac{\partial \mathbf{a}^{i+1}[T]}{\partial \mathbf{a}^i[T]} \frac{\partial \mathbf{a}^{l+1}[T]}{\partial \mathbf{W}^l}$, and we have $\frac{\partial L_{sr}}{\partial \hat{\mathbf{a}}^N[T]} = \frac{1}{\lambda^{T-t}} \frac{\partial L_{sr}}{\partial \mathbf{s}^N[t]} (\forall 1 \leq t \leq T)^2$, $\frac{\partial L_{sr}}{\partial \hat{\mathbf{a}}^N[T]} = \frac{1}{T} \sum_{t=1}^{T} \frac{1}{\lambda^{T-t}} \frac{\partial L_{sr}}{\partial \mathbf{s}^N[t]}$, we can obtain:

$$
\begin{aligned}
(\nabla_{\mathbf{W}^l} L_{sr})_{sr} = \left(\frac{\partial L_{sr}}{\partial \mathbf{W}^l}\right)_{sr}^{\top} &= \left(\frac{\partial L_{sr}}{\partial \mathbf{a}^N[T]} \prod_{i=N-1}^{l+1} \frac{\partial \mathbf{a}^{i+1}[T]}{\partial \mathbf{a}^i[T]} \frac{\partial \mathbf{a}^{l+1}[T]}{\partial \mathbf{W}^l}\right)^{\top} \\
&= \left(\left(\frac{\partial L_{sr}}{\partial \mathbf{a}^N[T]} \prod_{i=N-1}^{l+1} \frac{\partial \mathbf{a}^{i+1}[T]}{\partial \mathbf{a}^i[T]}\right)^{\top} \odot \mathbf{d}^{l+1}[T]\right) \mathbf{a}^l[T]^{\top} \\
&= \left(\left(\frac{\partial L_{sr}}{\partial \hat{\mathbf{a}}^N[T]} \prod_{i=N-1}^{l+1} \frac{\partial \mathbf{a}^{i+1}[T]}{\partial \mathbf{a}^i[T]}\right)^{\top} \odot \mathbf{d}^{l+1}[T]\right) \hat{\mathbf{a}}^l[T]^{\top} \\
&= \sum_{t=1}^{T} \left(\left(\frac{1}{T} \frac{1}{\lambda^{T-t}} \frac{\partial L_{sr}}{\partial \mathbf{s}^N[t]} \prod_{i=N-1}^{l+1} \frac{\partial \mathbf{a}^{i+1}[T]}{\partial \mathbf{a}^i[T]}\right)^{\top} \odot \mathbf{d}^{l+1}[T]\right) \hat{\mathbf{a}}^l[T]^{\top}.
\end{aligned}
\tag{1}
$$

### A.2 A few notes for the notation of time for multi-layer networks

Please note that the notation of discrete time steps for multi-layer networks may be slightly different from Eq. (2). We use $s^{i+1}[t]$ to denote the $(i+1)$-th layer's response after receiving the $i$-th layer's signals $s^i[t]$. Rigorously speaking, there will be synaptic delay $t_d$ for information propagation between two layers if we consider the whole network in an asynchronous way, so the precise time

---

[*]Corresponding author.

[2]Note that we can treat $\mathbf{s}^N[t]$ independent with each other, if we consider taking the derivative of the Heaviside step function as 0 in this calculation and therefore $\frac{\partial \mathbf{s}^N[t+1]}{\partial \mathbf{s}^N[t]} = \frac{\partial \mathbf{s}^N[t+1]}{\partial \mathbf{u}^N[t+1]} \frac{\partial \mathbf{u}^N[t+1]}{\partial \mathbf{s}^N[t]} = 0$.

36th Conference on Neural Information Processing Systems (NeurIPS 2022).

of $s^{i+1}[t]$ or $u^{i+1}[t]$ for layer $i+1$ may be $t + t_d$ compared with $s^i[t]$ for layer $i$. To simplify the notations, we use $0, 1, \cdots T$ for each layer to represent the corresponding discrete time steps, while the actual time of different layers at time step $t$ should consider some delay across layers.

### A.3 Proof of Theorem 1

In this subsection, we prove Theorem 1 with Assumption 1.

**Assumption 1.** $\forall l = 1, \cdots, N, t = 1, \cdots, T, \mathrm{diag}\left(\frac{\partial \mathbf{s}^{l+1}[t]}{\partial \mathbf{u}^{l+1}[t]}\right) = \mathbf{d}^{l+1}[\mathrm{T}].$

**Theorem 1.** *If Assumption 1 holds, $V_{th} = 1$, and the errors $\boldsymbol{\epsilon}^l[t] = \mathbf{a}^l[t] - \mathbf{a}^l[T]$ are small such that $\left\|\sum_{t=1}^{T} \hat{\mathbf{g}}_{\mathbf{u}^{l+1}}[t]\boldsymbol{\epsilon}^l[t]^\top\right\| < \left\|\sum_{t=1}^{T} \hat{\mathbf{g}}_{\mathbf{u}^{l+1}}[t]\mathbf{a}^l[T]^\top\right\| - \left\|\sum_{t=1}^{T} \frac{\lambda^t(1-\lambda^{T-t})}{1-\lambda^T}\hat{\mathbf{g}}_{\mathbf{u}^{l+1}}[t]\mathbf{a}^l[t]^\top\right\|$ when $(\nabla_{\mathbf{W}^l} L_{sr})_{sr} \neq \mathbf{0}$, then we have $\langle \nabla_{\mathbf{W}^l} L, (\nabla_{\mathbf{W}^l} L_{sr})_{sr} \rangle > 0.$*

*Proof.* As described in Sections 4.1 and 4.2, for gradients of OTTT, we have $\nabla_{\mathbf{W}^l} L = \sum_{t=1}^{T} \mathbf{g}_{\mathbf{u}^{l+1}}[t]\hat{\mathbf{a}}^l[t]^\top$, $L := \sum_{t=1}^{T} L[t] = \sum_{t=1}^{T} \frac{1}{T}\mathcal{L}\left(\mathbf{s}^N[t], \mathbf{y}\right)$, $\mathbf{g}_{\mathbf{u}^{l+1}}[t] = \left(\frac{\partial L[t]}{\partial \mathbf{s}^N[t]}\prod_{i=N-1}^{l+1}\frac{\partial \mathbf{s}^{i+1}[t]}{\partial \mathbf{s}^i[t]}\frac{\partial \mathbf{s}^{l+1}[t]}{\partial \mathbf{u}^{l+1}[t]}\right)^\top$; for gradients based on spike representation, we have $(\nabla_{\mathbf{W}^l} L_{sr})_{sr} = \sum_{t=1}^{T}\left(\left(\frac{1}{T}\frac{1}{\lambda^{T-t}}\frac{\partial L_{sr}}{\partial \mathbf{s}^N[t]}\prod_{i=N-1}^{l+1}\frac{\partial \mathbf{a}^{i+1}[T]}{\partial \mathbf{a}^i[T]}\right)^\top \odot \mathbf{d}^{l+1}[T]\right)\hat{\mathbf{a}}^l[T]^\top$ and consider

$L_{sr} = \frac{1}{\sum_{\tau=0}^{T-1}\lambda^\tau}\sum_{t=1}^{T}\lambda^{T-t}\mathcal{L}(\mathbf{s}^N[t], \mathbf{y})$. Let $\hat{\mathbf{g}}_{\mathbf{u}^{l+1}}[t] = \left(\frac{\partial \mathcal{L}(\mathbf{s}^N[t], \mathbf{y})}{\partial \mathbf{s}^N[t]}\prod_{i=N-1}^{l+1}\frac{\partial \mathbf{s}^{i+1}[t]}{\partial \mathbf{s}^i[t]}\frac{\partial \mathbf{s}^{l+1}[t]}{\partial \mathbf{u}^{l+1}[t]}\right)^\top$, we have $\nabla_{\mathbf{W}^l} L = \frac{1}{T}\sum_{t=1}^{T}\hat{\mathbf{g}}_{\mathbf{u}^{l+1}}[t]\hat{\mathbf{a}}^l[t]^\top$. With Assumption 1, we have $\frac{\partial \mathbf{s}_i^{l+1}[t]}{\partial \mathbf{u}_i^{l+1}[t]} = \mathbf{d}_i^{l+1}[T]$ and thus $\frac{\partial \mathbf{s}^{l+1}[t]}{\partial \mathbf{s}^l[t]} = \frac{\partial \mathbf{a}^{l+1}[t]}{\partial \mathbf{a}^l[t]}$ (because $\frac{\partial \mathbf{s}_j^{l+1}[t]}{\partial \mathbf{s}_i^l[t]} = \frac{\partial \mathbf{s}_j^{l+1}[t]}{\partial \mathbf{u}_j^{l+1}[t]} \cdot \mathbf{W}_{i,j} = \mathbf{d}_j^{l+1}[T] \cdot \mathbf{W}_{i,j} = \frac{\partial \mathbf{a}_j^{l+1}[t]}{\partial \mathbf{a}_i^l[t]}$). So we can derive that $(\nabla_{\mathbf{W}^l} L_{sr})_{sr} = \frac{1}{T}\frac{1}{\sum_{\tau=0}^{T-1}\lambda^\tau}\sum_{t=1}^{T}\hat{\mathbf{g}}_{\mathbf{u}^{l+1}}[t]\hat{\mathbf{a}}^l[T]^\top = \frac{1}{T}\sum_{t=1}^{T}\hat{\mathbf{g}}_{\mathbf{u}^{l+1}}[t]\mathbf{a}^l[T]^\top$.

We consider $\widehat{\nabla_{\mathbf{W}^l} L} = \frac{1}{\sum_{\tau=0}^{T-1}\lambda^\tau}\nabla_{\mathbf{W}^l} L = \frac{1}{T}\frac{1}{\sum_{\tau=0}^{T-1}\lambda^\tau}\sum_{t=1}^{T}\hat{\mathbf{g}}_{\mathbf{u}^{l+1}}[t]\hat{\mathbf{a}}^l[t]^\top = \frac{1}{T}\sum_{t=1}^{T}\hat{\mathbf{g}}_{\mathbf{u}^{l+1}}[t]\frac{\sum_{\tau=0}^{t-1}\lambda^\tau}{\sum_{\tau=0}^{T-1}\lambda^\tau}\mathbf{a}^l[t]^\top$. Since the errors $\boldsymbol{\epsilon}^l[t] = \mathbf{a}^l[t] - \mathbf{a}^l[T]$ are small such that $\left\|\sum_{t=1}^{T}\hat{\mathbf{g}}_{\mathbf{u}^{l+1}}[t]\boldsymbol{\epsilon}^l[t]^\top\right\| < \left\|\sum_{t=1}^{T}\hat{\mathbf{g}}_{\mathbf{u}^{l+1}}[t]\mathbf{a}^l[T]^\top\right\| - \left\|\sum_{t=1}^{T}\frac{\lambda^t(1-\lambda^{T-t})}{1-\lambda^T}\hat{\mathbf{g}}_{\mathbf{u}^{l+1}}[t]\mathbf{a}^l[t]^\top\right\|$ when $(\nabla_{\mathbf{W}^l} L_{sr})_{sr} \neq \mathbf{0}$, we have:

$$\left\|\widehat{\nabla_{\mathbf{W}^l} L} - (\nabla_{\mathbf{W}^l} L_{sr})_{sr}\right\| = \left\|\frac{1}{T}\sum_{t=1}^{T}\hat{\mathbf{g}}_{\mathbf{u}^{l+1}}[t]\left(\frac{\sum_{\tau=0}^{t-1}\lambda^\tau}{\sum_{\tau=0}^{T-1}\lambda^\tau}\mathbf{a}^l[t]^\top - \mathbf{a}^l[T]^\top\right)\right\|$$

$$= \left\|\frac{1}{T}\sum_{t=1}^{T}\hat{\mathbf{g}}_{\mathbf{u}^{l+1}}[t]\left(\boldsymbol{\epsilon}^l[t] - \frac{\lambda^t(1-\lambda^{T-t})}{1-\lambda^T}\mathbf{a}^l[t]^\top\right)\right\|$$

$$\leq \left\|\frac{1}{T}\sum_{t=1}^{T}\hat{\mathbf{g}}_{\mathbf{u}^{l+1}}[t]\boldsymbol{\epsilon}^l[t]\right\| + \left\|\frac{1}{T}\sum_{t=1}^{T}\hat{\mathbf{g}}_{\mathbf{u}^{l+1}}[t]\frac{\lambda^t(1-\lambda^{T-t})}{1-\lambda^T}\mathbf{a}^l[t]^\top\right\| \tag{2}$$

$$< \left\|\frac{1}{T}\sum_{t=1}^{T}\hat{\mathbf{g}}_{\mathbf{u}^{l+1}}[t]\mathbf{a}^l[T]^\top\right\| = \left\|(\nabla_{\mathbf{W}^l} L_{sr})_{sr}\right\|.$$

Then, we can obtain:

$$\left\langle \widehat{\nabla_{\mathbf{W}^l} L}, (\nabla_{\mathbf{W}^l} L_{sr})_{sr}\right\rangle = \left\langle \widehat{\nabla_{\mathbf{W}^l} L} - (\nabla_{\mathbf{W}^l} L_{sr})_{sr}, (\nabla_{\mathbf{W}^l} L_{sr})_{sr}\right\rangle + \left\|(\nabla_{\mathbf{W}^l} L_{sr})_{sr}\right\|^2$$

$$\geq \left\|(\nabla_{\mathbf{W}^l} L_{sr})_{sr}\right\|^2 - \left\|\widehat{\nabla_{\mathbf{W}^l} L} - (\nabla_{\mathbf{W}^l} L_{sr})_{sr}\right\|\left\|(\nabla_{\mathbf{W}^l} L_{sr})_{sr}\right\| > 0. \tag{3}$$

Therefore, $\langle \nabla_{\mathbf{W}^l} L, (\nabla_{\mathbf{W}^l} L_{sr})_{sr}\rangle = \left(\sum_{\tau=0}^{T-1}\lambda^\tau\right)\left\langle \widehat{\nabla_{\mathbf{W}^l} L}, (\nabla_{\mathbf{W}^l} L_{sr})_{sr}\right\rangle > 0.$

$\square$

**Remark 1.** *As for the assumption of the errors in the theorem, since the weighted firing rate gradually converges $\mathbf{a}[t] \to \mathbf{a}^*$ with bounded random error caused by the remaining membrane potential at the last time step, the order of errors $\boldsymbol{\epsilon}^l[t]$ would be smaller than $\mathbf{a}^l[T]$ especially when $t$ is large. And $\frac{\lambda^t(1-\lambda^{T-t})}{1-\lambda^T} \to 0$ with $t \to T$ is also a small number on the right side of the inequality. So this is a reasonable assumption.*

**Remark 2.** *The above conclusion mainly focuses on the gradients for connection weights $\mathbf{W}^l$. As for other parameters such as biases $\mathbf{b}^l$, the gradients of OTTT do not involve pre-synaptic activities, so under Assumption 1 they are exactly the same as gradients based on spike representation except a constant scaling factor $\frac{1}{\sum_{\tau=0}^{T-1} \lambda^\tau}$.*

**Remark 3.** *Note that the gradients based on spike representation may also include small errors since the calculation of SNN is not exactly the same as the equivalent ANN-like mappings. And a larger time step may lead to more accurate gradients. We connect the gradients of OTTT and gradients based on spike representation to demonstrate the overall descent direction, and it is tolerant to small errors, which can also be viewed as randomness for stochastic optimization.*

### A.4 Proof of Theorem 2

In this subsection, we prove Theorem 2.

**Theorem 2.** *If Assumption 1 holds, $V_{th} = 1$, $\left\| J_{f_{\boldsymbol{\theta}}} |_{\mathbf{a}[T]} \right\| \leq \eta < \frac{\sigma_{min}^2}{\sigma_{max}^2}$, where $\sigma_{max}$ and $\sigma_{min}$ are the maximal and minimal singular value of $\frac{\partial f_{\boldsymbol{\theta}}}{\partial \boldsymbol{\theta}} |_{\mathbf{a}[T]}$, and the errors $\boldsymbol{\epsilon}^1[t] = \mathbf{a}[t] - \mathbf{a}[T], \boldsymbol{\epsilon}^0[t] = \overline{\mathbf{x}}[t] - \overline{\mathbf{x}}[T]$ are small such that $\left\| \sum_{t=1}^{T} \hat{\mathbf{g}}_{\mathbf{u}}[t] \boldsymbol{\epsilon}^l[t]^\top \right\| < \frac{\sigma_{min}^2 - \eta \sigma_{max}^2}{\sigma_{max}} \left\| \sum_{t=1}^{T} \frac{\partial \mathcal{L}(\mathbf{s}[t], \mathbf{y})}{\partial \mathbf{s}[t]} \left(I - J_{f_{\boldsymbol{\theta}}} |_{\mathbf{a}[T]}\right)^{-1} \right\| - \left\| \sum_{t=1}^{T} \frac{\lambda^t(1-\lambda^{T-t})}{1-\lambda^T} \hat{\mathbf{g}}_{\mathbf{u}}[t] \mathbf{a}^l[t]^\top \right\|$ (where $l = 0, 1$, $\mathbf{a}^1[t]$ and $\mathbf{a}^0[t]$ represent $\mathbf{a}[t]$ and $\overline{\mathbf{x}}[t]$, respectively) when $(\nabla_{\boldsymbol{\theta}} L_{sr})_{sr} \neq \mathbf{0}$, then we have $\langle \nabla_{\boldsymbol{\theta}} L, (\nabla_{\boldsymbol{\theta}} L_{sr})_{sr} \rangle > 0$, where $\boldsymbol{\theta}$ are parameters in the network.*

*Proof.* As described in Sections 4.1 and 4.2 and similar to the proof of Theorem 1, let $\hat{\mathbf{g}}_{\mathbf{u}}[t] = \left( \frac{\partial \mathcal{L}(\mathbf{s}[t], \mathbf{y})}{\partial \mathbf{s}[t]} \frac{\partial \mathbf{s}[t]}{\partial \mathbf{u}[t]} \right)^\top$, we have $\nabla_{\mathbf{W}} L = \frac{1}{T} \sum_{t=1}^{T} \hat{\mathbf{g}}_{\mathbf{u}}[t] \hat{\mathbf{a}}[t]^\top$, $\nabla_{\mathbf{F}} L = \frac{1}{T} \sum_{t=1}^{T} \hat{\mathbf{g}}_{\mathbf{u}}[t] \hat{\mathbf{x}}[t]^\top$ (where $\hat{\mathbf{x}}[t] = \sum_{\tau=1}^{t} \lambda^{t-\tau} \mathbf{x}[\tau]$), and $\nabla_{\mathbf{b}} L = \frac{1}{T} \sum_{t=1}^{T} \hat{\mathbf{g}}_{\mathbf{u}}[t]$. For gradients based on spike representation, $(\nabla_{\boldsymbol{\theta}} L_{sr})_{sr} = \left( \frac{\partial L_{sr}}{\partial \mathbf{a}[T]} \left(I - J_{f_{\boldsymbol{\theta}}} |_{\mathbf{a}[T]}\right)^{-1} \frac{\partial f_{\boldsymbol{\theta}}(\mathbf{a}[T])}{\partial \boldsymbol{\theta}} \right)^\top$, and we will also consider $(\widetilde{\nabla_{\boldsymbol{\theta}} L_{sr}})_{sr} = \left( \frac{\partial L_{sr}}{\partial \mathbf{a}[T]} \frac{\partial f_{\boldsymbol{\theta}}(\mathbf{a}[T])}{\partial \boldsymbol{\theta}} \right)^\top$. Considering $L_{sr} = \frac{1}{\sum_{\tau=0}^{T-1} \lambda^\tau} \sum_{t=1}^{T} \lambda^{T-t} \mathcal{L}(\mathbf{s}[t], \mathbf{y})$, and with Assumption 1 which indicates $\frac{\partial \mathbf{s}_i^{l+1}[t]}{\partial \mathbf{u}_i^{l+1}[t]} = \mathbf{d}_i^{l+1}[T]$, we can derive that $(\widetilde{\nabla_{\mathbf{W}} L_{sr}})_{sr} = \frac{1}{T} \sum_{t=1}^{T} \hat{\mathbf{g}}_{\mathbf{u}}[t] \mathbf{a}[T]^\top$, $(\widetilde{\nabla_{\mathbf{F}} L_{sr}})_{sr} = \frac{1}{T} \sum_{t=1}^{T} \hat{\mathbf{g}}_{\mathbf{u}}[t] \overline{\mathbf{x}}[T]^\top$, and $(\widetilde{\nabla_{\mathbf{b}} L_{sr}})_{sr} = \frac{1}{T} \frac{1}{\sum_{\tau=0}^{T-1} \lambda^\tau} \sum_{t=1}^{T} \hat{\mathbf{g}}_{\mathbf{u}}[t]$.

We consider $\widehat{\nabla_{\mathbf{W}} L} = \frac{1}{\sum_{\tau=0}^{T-1} \lambda^\tau} \nabla_{\mathbf{W}} L = \frac{1}{T} \sum_{t=1}^{T} \hat{\mathbf{g}}_{\mathbf{u}^{l+1}}[t] \frac{\sum_{\tau=0}^{t-1} \lambda^\tau}{\sum_{\tau=0}^{T-1} \lambda^\tau} \mathbf{a}^l[t]^\top$ and $\widehat{\nabla_{\mathbf{F}} L} = \frac{1}{\sum_{\tau=0}^{T-1} \lambda^\tau} \nabla_{\mathbf{F}} L$. Since $\left\| J_{f_{\boldsymbol{\theta}}} |_{\mathbf{a}[T]} \right\| \leq \eta < \frac{\sigma_{min}^2}{\sigma_{max}^2}$, where $\sigma_{max}$ and $\sigma_{min}$ are the maximal and minimal singular value of $\frac{\partial f_{\boldsymbol{\theta}}}{\partial \boldsymbol{\theta}} |_{\mathbf{a}[T]}$ ($\boldsymbol{\theta} \in \{\mathbf{W}, \mathbf{F}, \mathbf{b}\}$), and the errors $\boldsymbol{\epsilon}[t] = \mathbf{a}[t] - \mathbf{a}[T]$ are small such that $\left\| \sum_{t=1}^{T} \hat{\mathbf{g}}_{\mathbf{u}}[t] \boldsymbol{\epsilon}^l[t]^\top \right\| < \frac{\sigma_{min}^2 - \eta \sigma_{max}^2}{\sigma_{max}} \left\| \sum_{t=1}^{T} \frac{\partial \mathcal{L}(\mathbf{s}[t], \mathbf{y})}{\partial \mathbf{s}[t]} \left(I - J_{f_{\boldsymbol{\theta}}} |_{\mathbf{a}[T]}\right)^{-1} \right\| - \left\| \sum_{t=1}^{T} \frac{\lambda^t(1-\lambda^{T-t})}{1-\lambda^T} \hat{\mathbf{g}}_{\mathbf{u}}[t] \mathbf{a}[t]^\top \right\|$ when $(\nabla_{\boldsymbol{\theta}} L)_{sr} \neq \mathbf{0}$, we can obtain:

$$
\begin{aligned}
\left\| \widehat{\nabla_{\mathbf{W}} L} - (\widetilde{\nabla_{\mathbf{W}} L_{sr}})_{sr} \right\| &= \left\| \frac{1}{T} \sum_{t=1}^{T} \hat{\mathbf{g}}_{\mathbf{u}}[t] \left( \frac{\sum_{\tau=0}^{t-1} \lambda^\tau}{\sum_{\tau=0}^{T-1} \lambda^\tau} \mathbf{a}[t]^\top - \mathbf{a}[T]^\top \right) \right\| \\
&= \left\| \frac{1}{T} \sum_{t=1}^{T} \hat{\mathbf{g}}_{\mathbf{u}}[t] \left( \boldsymbol{\epsilon}[t] - \frac{\lambda^t(1-\lambda^{T-t})}{1-\lambda^T} \mathbf{a}[t]^\top \right) \right\| \\
&\leq \left\| \frac{1}{T} \sum_{t=1}^{T} \hat{\mathbf{g}}_{\mathbf{u}}[t] \boldsymbol{\epsilon}[t] \right\| + \left\| \frac{1}{T} \sum_{t=1}^{T} \hat{\mathbf{g}}_{\mathbf{u}}[t] \frac{\lambda^t(1-\lambda^{T-t})}{1-\lambda^T} \mathbf{a}[t]^\top \right\|
\end{aligned}
$$

$$< \frac{\sigma_{\min}^2 - \eta\sigma_{\max}^2}{\sigma_{\max}} \left\| \frac{1}{T}\sum_{t=1}^{T} \frac{\partial \mathcal{L}(\mathbf{s}[t],\mathbf{y})}{\partial \mathbf{s}[t]} \left(I - J_{f_{\boldsymbol{\theta}}}|_{\mathbf{a}[T]}\right)^{-1} \right\|. \qquad (4)$$

Then, we have (let $\mathbf{v} = \left(\frac{\partial L_{sr}}{\partial \mathbf{a}[T]}\left(I - J_{f_{\boldsymbol{\theta}}}|_{\mathbf{a}[T]}\right)^{-1}\right)^{\top} = \frac{1}{T}\sum_{t=1}^{T}\left(\frac{\partial \mathcal{L}(\mathbf{s}[t],\mathbf{y})}{\partial \mathbf{s}[t]}\left(I - J_{f_{\boldsymbol{\theta}}}|_{\mathbf{a}[T]}\right)^{-1}\right)^{\top}$):

$$
\begin{aligned}
\left\langle \widehat{\nabla_{\mathbf{W}}L}, (\nabla_{\mathbf{W}}L_{sr})_{sr} \right\rangle &= \left\langle (\widetilde{\nabla_{\mathbf{W}}L_{sr}})_{sr}, (\nabla_{\mathbf{W}}L_{sr})_{sr} \right\rangle + \left\langle \widehat{\nabla_{\mathbf{W}}L} - (\widetilde{\nabla_{\mathbf{W}}L_{sr}})_{sr}, (\nabla_{\mathbf{W}}L_{sr})_{sr} \right\rangle \\
&= \mathbf{v}^{\top}\frac{\partial f_{\boldsymbol{\theta}}(\mathbf{a}[T])}{\partial \mathbf{W}}\left(\frac{\partial L_{sr}}{\partial \mathbf{a}[T]}\frac{\partial f_{\boldsymbol{\theta}}(\mathbf{a}[T])}{\partial \mathbf{W}}\right)^{\top} + \left\langle \widehat{\nabla_{\mathbf{W}}L} - (\widetilde{\nabla_{\mathbf{W}}L_{sr}})_{sr}, (\nabla_{\mathbf{W}}L_{sr})_{sr} \right\rangle \\
&= \mathbf{v}^{\top}\frac{\partial f_{\boldsymbol{\theta}}(\mathbf{a}[T])}{\partial \mathbf{W}}\frac{\partial f_{\boldsymbol{\theta}}(\mathbf{a}[T])}{\partial \mathbf{W}}^{\top}\left(I - J_{f_{\boldsymbol{\theta}}}|_{\mathbf{a}[T]}\right)^{\top}\mathbf{v} + \left\langle \widehat{\nabla_{\mathbf{W}}L} - (\widetilde{\nabla_{\mathbf{W}}L_{sr}})_{sr}, (\nabla_{\mathbf{W}}L_{sr})_{sr} \right\rangle \\
&= \left\| \mathbf{v}^{\top}\frac{\partial f_{\boldsymbol{\theta}}(\mathbf{a}[T])}{\partial \mathbf{W}} \right\|^2 - \mathbf{v}^{\top}\frac{\partial f_{\boldsymbol{\theta}}(\mathbf{a}[T])}{\partial \mathbf{W}}\frac{\partial f_{\boldsymbol{\theta}}(\mathbf{a}[T])}{\partial \mathbf{W}}^{\top}J_{f_{\boldsymbol{\theta}}}|_{\mathbf{a}[T]}^{\top}\mathbf{v} \\
&\quad + \left\langle \widehat{\nabla_{\mathbf{W}}L} - (\widetilde{\nabla_{\mathbf{W}}L_{sr}})_{sr}, (\nabla_{\mathbf{W}}L_{sr})_{sr} \right\rangle \\
&\geq \sigma_{\min}^2\|\mathbf{v}\|^2 - \eta\sigma_{\max}^2\|\mathbf{v}\|^2 - \left\| \widehat{\nabla_{\mathbf{W}}L} - (\widetilde{\nabla_{\mathbf{W}}L_{sr}})_{sr} \right\| \left\| \mathbf{v}^{\top}\frac{\partial f_{\boldsymbol{\theta}}(\mathbf{a}[T])}{\partial \mathbf{W}} \right\| \\
&> \sigma_{\min}^2\|\mathbf{v}\|^2 - \eta\sigma_{\max}^2\|\mathbf{v}\|^2 - \frac{\sigma_{\min}^2 - \eta\sigma_{\max}^2}{\sigma_{\max}}\|\mathbf{v}\|\cdot\sigma_{\max}\|\mathbf{v}\| = 0. \qquad (5)
\end{aligned}
$$

Therefore, $\left\langle \nabla_{\mathbf{W}}L, (\nabla_{\mathbf{W}}L_{sr})_{sr} \right\rangle = \left(\sum_{\tau=0}^{T-1}\lambda^{\tau}\right)\left\langle \widehat{\nabla_{\mathbf{W}}L}, (\nabla_{\mathbf{W}}L_{sr})_{sr} \right\rangle > 0$. Similarly, we can derive that $\langle \nabla_{\mathbf{F}}L, (\nabla_{\mathbf{F}}L_{sr})_{sr} \rangle > 0$. And for $\nabla_{\mathbf{b}}L$, we have $\nabla_{\mathbf{b}}L = \left(\sum_{\tau=0}^{T-1}\lambda^{\tau}\right)(\nabla_{\mathbf{b}}L_{sr})_{sr}$, so $\langle \nabla_{\mathbf{b}}L, (\nabla_{\mathbf{b}}L_{sr})_{sr} \rangle > 0$. Therefore, for all parameters $\boldsymbol{\theta}$ in the network, we have $\langle \nabla_{\boldsymbol{\theta}}L, (\nabla_{\boldsymbol{\theta}}L_{sr})_{sr} \rangle > 0$ when $(\nabla_{\boldsymbol{\theta}}L_{sr})_{sr} \neq \mathbf{0}$.

$\square$

**Remark 4.** *The above conclusion considers the single-layer condition. It can be generalized to the multi-layer condition. For example, if we consider multiple feedforward hidden layers (denote the weight as $F^l$) with a feedback connection from the last hidden layer to the first hidden layer (denote the weight as $W^1$), and assume the function is contractive, the equilibrium states for each layer are $\mathbf{a}^{1^*} = f_1\left(f_N \circ \cdots \circ f_2(\mathbf{a}^{1^*}), \mathbf{x}^*\right)$ and $\mathbf{a}^{l+1^*} = f_{l+1}(\mathbf{a}^{l^*})$, where $f_1(\mathbf{a},\mathbf{x}) = \sigma\left(\frac{1}{V_{th}}(\mathbf{W}^1\mathbf{a} + \mathbf{F}^1\mathbf{x} + \mathbf{b}^1)\right)$ and $f_l(\mathbf{a}) = \sigma\left(\frac{1}{V_{th}}(\mathbf{F}^l\mathbf{a} + \mathbf{b}^l)\right)$ [1]. Then with a similar condition for the Jacobian of $f_{\boldsymbol{\theta}} = f_N \circ \cdots \circ f_2 \circ f_1$ and errors $\boldsymbol{\epsilon}^l[t]$ of each layer as in Theorem 2, we can prove $\langle \nabla_{\boldsymbol{\theta}}L, (\nabla_{\boldsymbol{\theta}}L)_{sr} \rangle > 0$ when $(\nabla_{\boldsymbol{\theta}}L)_{sr} \neq \mathbf{0}$ for all parameters $\boldsymbol{\theta}$ in the network as well. More generally, multi-layer networks with arbitrary feedback connections can be written in a single-layer formulation, i.e. we consider all neurons in different layers as a whole single layer, and feedforward or feedback connections can be viewed as connections between these neurons, which is written as a much larger weight matrix with some imposed structures representing the connection restrictions. Therefore, the conclusion can be directly generalized to these conditions as well.*

**Remark 5.** *The assumption $\left\|J_{f_{\boldsymbol{\theta}}}|_{\mathbf{a}[T]}\right\| \leq \eta < \frac{\sigma_{min}^2}{\sigma_{max}^2}$ is also made in previous works [2, 3] and we consider it as a reasonable assumption in the theoretical analysis. It is a sufficient condition to bound the worst case, and in practice it is unnecessary to always enforce the restriction, as indicated in [2].*

## B  Pseudocode of the OTTT algorithm

We present the pseudocode of one iteration of OTTT training for a feedforward network in Algorithm 1 to better illustrate our training method.

---

**Algorithm 1** One iteration of OTTT training for a feedforward network.

---

**Input:** Network parameters $\{\mathbf{W}^l\}$, $\{\mathbf{b}^l\}$; Input data $x$; Label $y$; Time steps $T$; Other hyperparameters;

**Output:** Trained network parameters $\{\mathbf{W}^l\}$, $\{\mathbf{b}^l\}$.

 1: **for** $t = 1, 2, \cdots, T$ **do**
 2:       **for** $l = 1, 2, \cdots, N$ **do**     // **Forward**
 3:             Update membrane potentials $\mathbf{u}^l[t]$ and generate spikes $\mathbf{s}^l[t]$ at layer $l$;
 4:             Update the tracked presynaptic activities $\hat{\mathbf{a}}^l[t] = \lambda\hat{\mathbf{a}}^l[t-1] + \hat{\mathbf{s}}^l[t]$ at layer $l$.
 5:       **for** $l = N, N-1, \cdots, 1$ **do**     // **Backward**
 6:             Calculate the instantaneous backpropagated errors $\mathbf{g}_{\mathbf{u}^l}[t]$;
 7:             Calculate the instantaneous gradient $\nabla_{\mathbf{W}^{l-1}}L[t] = \mathbf{g}_{\mathbf{u}^l}[t](\hat{\mathbf{a}}^{l-1}[t])^\top$.
 8:             **if** online update **then**     // OTTT$_O$
 9:                   Update $\mathbf{W}^{l-1}$ with $\nabla_{\mathbf{W}^{l-1}}L[t]$ based on the gradient-based optimizer;
10:                   Update $\mathbf{b}^l$ with $\mathbf{g}_{\mathbf{u}^l}[t]$ based on the gradient-based optimizer.
11:             **else**     // OTTT$_A$
12:                   Accumulate gradients $\nabla_{\mathbf{W}^{l-1}}L = \nabla_{\mathbf{W}^{l-1}}L + \nabla_{\mathbf{W}^{l-1}}L[t]$, $\nabla_{\mathbf{b}^l}L = \nabla_{\mathbf{b}^l}L + \mathbf{g}_{\mathbf{u}^l}[t]$.
13: **if** not online update **then**     // OTTT$_A$
14:       Update parameters $\{\mathbf{W}^l\}$ with accumulated $\{\nabla_{\mathbf{W}^l}L\}$ based on the gradient-based optimizer;
15:       Update parameters $\{\mathbf{b}^l\}$ with accumulated $\{\nabla_{\mathbf{b}^l}L\}$ based on the gradient-based optimizer.

## C   Implementation Details

### C.1   Scaled Weight Standardization and NF-ResNets

The scaled weight standardization (sWS) is proposed in [4, 5] to replace the commonly used batch normalization (BN) and realize normalization-free ResNets (NF-ResNets). Different from BN which standardizes the activation with different samples, sWS standardizes weights by:

$$\hat{\mathbf{W}}_{i,j} = \gamma \cdot \frac{\mathbf{W}_{i,j} - \mu_{\mathbf{W}_{i,\cdot}}}{\sigma_{\mathbf{W}_{i,\cdot}}\sqrt{N}}, \tag{6}$$

where $\mu_{\mathbf{W}_{i,\cdot}}$ and $\sigma_{\mathbf{W}_{i,\cdot}}$ are the mean and variance calculated along the input dimension, and the scale $\gamma$ is determined by analyzing the signal propagation with different activation functions. The original weight standardization is proposed in [6], which is shown to share the similar benefit as BN to smooth the loss landscape, if combined with other normalization techniques, e.g. group normalization. sWS further takes the signal propagation into account so that the variance of the signal is preserved during the forward propagation of neural networks and the mean of the output is 0, which is another property of BN. Particularly, for the input $\mathbf{x}$ that is sampled i.i.d from $\mathcal{N}(0, 1)$, considering the ReLU activation $g$, [4] derive that we should take $\gamma = \frac{\sqrt{2}}{\sqrt{1-\frac{1}{\pi}}}$ to preserve the variance of signals, i.e. $\mathrm{Var}(\hat{\mathbf{W}}g(\mathbf{x})) = 1$. This is because the outputs $g(x) = \max(x, 0)$ with Gaussian inputs will be sampled from the rectified Gaussian distribution with variance $\sigma_g^2 = (1/2)(1 - (1/\pi))$ [4]. In this work, to ensure the variance preserving at each time step of the SNN computation, we derive $\gamma$ based on the consideration of the signals after the Heaviside step function $H$. Particularly, consider the Gaussian input $\mathbf{x}$, when $V_{th} = 1$, the variance of the outputs $H(x - V_{th})$ is $\sigma_H^2 = \frac{1}{2}\mathrm{erfc}(\frac{1}{\sqrt{2}})\left(1 - \frac{1}{2}\mathrm{erfc}(\frac{1}{\sqrt{2}})\right)$. So we will take $\gamma = \frac{1}{\sigma_H} \approx 2.74$ to preserve the variance of signals. Additionally, [4] demonstrates that sWS can incorporate another learnable scaling factor for the weights, which is also taken in common BN implementations. Therefore, we also adopt this sWS technique, which is the same as the pseudocode in [4]. For VGG network structures, we directly impose sWS on all weights. For NF-ResNet structures, we use the same structure as in [4], which is briefly introduced below.

NF-ResNets [4] consider the residual networks $x_{l+1} = x_l + \alpha f_l(x_l/\beta_l)$, which differs from ResNets [7] in three aspects: 1. NF-ResNets remove the BN components in ResNets and impose sWS on all weights; 2. a scaling factor $\alpha$ is added for each residual branch; 3. for the input of each residual branch, it will first be divided by the term $\beta_l$ that represents the standard deviation of signals. Note that the third point is because the residual computation $x_{l+1} = x_l + \alpha f_l(x_l/\beta_l)$ will gradually

accumulate the variance of the residual branch, i.e. $\text{Var}(x_{l+1}) = \text{Var}(x_l) + \text{Var}(\alpha f_l(x_l/\beta_l))$, so dividing $\beta_l$ ensures that the residual branch keeps the identity variance 1 (combined with sWS), and this also indicates to calculate $\beta_l$ by $\beta_{l+1}^2 = \beta_l^2 + \alpha^2$ after each branch. Also, note that for each transition block, the identity path with a strided conv will also be first divided by $\beta_l$, so the variance is reset after each transition block between two stages. For the implementation details, we mainly follow the pseudocode in [4] and replace the activation functions by functions of spiking neurons, and we take $\alpha = 0.2$. For more illustrations and other details, please directly refer to [4].

## C.2 Training Settings

### C.2.1 Datasets

We conduct experiments on CIFAR-10 [8], CIFAR-100 [8], ImageNet [9], CIFAR10-DVS [10], and DVS128-Gesture [11].

**CIFAR-10** CIFAR-10 is a dataset of color images with 10 classes of objects, which contains 50,000 training samples and 10,000 testing samples. Each sample is a $32 \times 32 \times 3$ color image. We normalize the inputs based on the global mean and standard deviation, and apply random cropping, horizontal flipping and cutout [12] for data augmentation. The inputs to the first layer of SNNs at each time step are directly the pixel values, which can be viewed as a real-valued input current.

**CIFAR-100** CIFAR-100 is a dataset similar to CIFAR-10 except that there are 100 classes of objects. It also consists of 50,000 training samples and 10,000 testing samples. We use the same pre-processing as CIFAR-10.

The license of CIFAR-10 and CIFAR-100 is the MIT License.

**ImageNet** ImageNet-1K is a dataset of color images with 1000 classes of objects, which contains 1,281,167 training samples and 50,000 validation images. We adopt the common pre-possessing strategies, i.e. the training images are first randomly resized and cropped to $224 \times 224$, and then normalized after the random horizontal flipping data augmentation, while the testing images are first resized to $256 \times 256$ and center-cropped to $224 \times 224$, and then normalized. The inputs are also converted to a real-valued input current at each time step. The license of ImageNet is Custom (non-commercial).

**DVS-CIFAR10** The DVS-CIFAR10 dataset is the neuromorphic version of the CIFAR-10 dataset converted by a Dynamic Vision Sensor (DVS), which is composed of 10,000 samples, one-sixth of the original CIFAR-10. It consists of spike trains with two channels corresponding to ON- and OFF-event spikes. The pixel dimension is expanded to $128 \times 128$. Following the common practice, we split the dataset into 9000 training samples and 1000 testing samples. As for the data pre-processing, we reduce the time resolution by accumulating the spike events [13] into 10 time steps, and we reduce the spatial resolution into $48 \times 48$ by interpolation. We apply the random cropping augmentation as CIFAR-10 to the input data, and normalize the inputs based on the global mean and standard deviation of all time steps (which can be integrated into the connection weights of the first layer). The license of DVS-CIFAR10 is CC BY 4.0.

**DVS128-Gesture** The DVS128-Gesture dataset is a neuromorphic dataset that contains 11 kinds of hand gestures from 29 subjects under 3 kinds of illumination conditions recorded by a DVS camera. It is composed of 1176 training samples and 288 testing samples. Following [13], we pre-possess the data to integrate event data into 20 frames. The license of DVS128-Gesture is the Creative Commons Attribution 4.0 license.

### C.2.2 Training Hyperparameters

For our SNN models, we assume the neurons of the last classification layer will not spike or reset, and do classification based on the accumulated membrane potential, which is the same as [1]. That is, the final output is $\mathbf{u}^N[t] = \mathbf{W}^{N-1}\mathbf{s}^{N-1}[t] + \mathbf{b}^N$ at each time step. The classification is based on the accumulated $\mathbf{u}^N = \sum_{t=1}^{T} \mathbf{u}^N[t]$, and the loss during training is also calculated based on $\mathbf{u}^N[t]$, i.e. $\mathcal{L}(\mathbf{u}^N[t], \mathbf{y})$.

For CIFAR-10, CIFAR-100, and DVS-CIFAR10, models are trained by SGD with momentum 0.9 for 300 epochs with the default batch size 128, and the initial learning rate is set as 0.1 with a cosine annealing learning rate scheduler to 0 (for the experiments of training with batch size 1, the initial learning rate is linearly rescaled to $\frac{0.1}{128}$). For DVS-CIFAR10, we apply dropout on all layers with dropout rate as 0.1. As for the loss function, inspired by [14], we combine cross-entropy (CE) loss and mean-square-error (MSE) loss, i.e. $\mathcal{L}(\mathbf{u}^N[t], \mathbf{y}) = (1 - \alpha)\mathbf{CE}(\mathbf{u}^N[t], \mathbf{y}) + \alpha\mathbf{MSE}(\mathbf{u}^N[t], \mathbf{y})$, where $\alpha$ is taken as 0.05 for CIFAR10 and CIFAR100 while 0.001 for DVS-CIFAR10.

For ImageNet, models are trained by SGD with momentum 0.9 for 100 epochs with the default batch size 256, and the initial learning rate is set as 0.1, which is decayed by 0.1 every 30 epochs. We set the weight decay as $2 \times 10^{-5}$, and no dropout is applied. The loss function takes the cross-entropy loss.

For DVS128-Gesture, models are trained by the Adam optimizer for 300 epochs with batch size 16, and the initial learning rate is set as 0.001 with a cosine annealing learning rate scheduler to 0. No dropout is applied. As for the loss function, we set $\alpha = 0.001$ following DVS-CIFAR10.

The code implementation is based on the PyTorch framework [15], and experiments are carried out on one NVIDIA GeForce RTX 3090 GPU.

# D   Additional Experiment Results

## D.1   Firing Rate Statistics on ImageNet

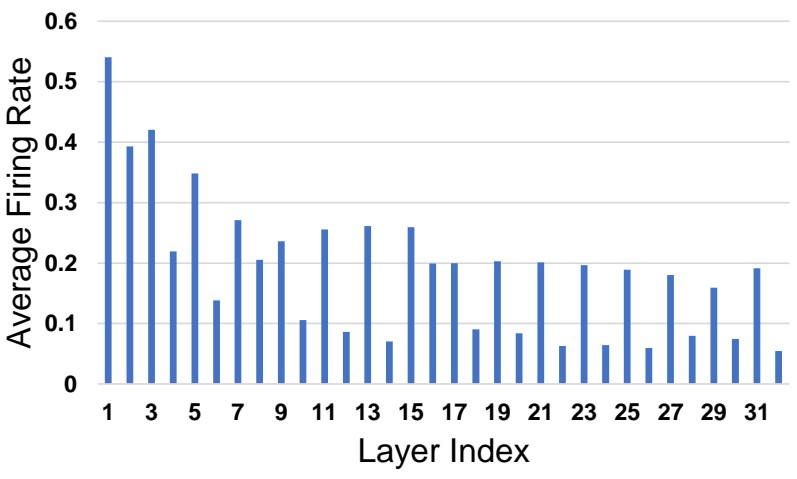

Figure 1: The average firing rates for the model trained by $\text{OTTT}_A$ on ImageNet.

In this section, we supplement the firing rate statistics of the NF-ResNet-34 model trained by $\text{OTTT}_A$ on ImageNet, as shown in Fig. 1. Overall the firing rate is around 0.24 and with 6 time steps each neuron averagely generate 1.46 spikes. Note that we can also reduce the time steps to realize a trade-off between accuracy and energy, as shown in Fig. 3 in Section 5.5. For example, with 2 time steps each neuron only averagely generate 0.48 spikes, with around 2.5% accuracy drop.

## D.2   Comparison between OTTT and BPTT with Feedback Connections

In this section, we supplement the results to compare the performance of OTTT and BPTT with feedback connections. As shown in Table 1, feedback connections can improve the performance for both OTTT and BPTT, and the improvement of OTTT from feedback connections is more significant than that of BPTT.

Table 1: Performance on CIFAR-100 for VGG and VGG-F trained by OTTT$_O$ and BPTT.

| Method | Network structure | Params | Mean±Std (Best) |
|---|---|---|---|
| OTTT$_O$ (ours) | VGG | 9.3M | 71.05±0.06% (71.11%) |
| OTTT$_O$ (ours) | VGG-F | 9.6M | 72.63±0.23% (72.94%) |
| BPTT | VGG | 9.3M | 69.06±0.07% (69.15%) |
| BPTT | VGG-F | 9.6M | (69.49%) |

### D.3 Experiments on Fully Recurrent Structures

In this section, we supplement an experiment to use a recurrent spiking neural network to classify the Fashion-MNIST dataset [16]. The input is flattened as a vector with 784 dimensions, and is connected to 400 spiking neurons with recurrent connections, which are then connected to a readout layer for classification. We apply weight standardization for connection weights from inputs to hidden neurons. Models are trained by 100 epochs with batch size 128 and SGD with momentum 0.9. The initial learning rate is set as 0.1 with a cosine annealing learning rate scheduler to 0. Dropout is set as 0.2, and weight decay is set as 5e-4 for BPTT and OTTT$_A$ while 1e-4 for OTTT$_O$ (since OTTT$_O$ update more times for each iteration). As for the loss function, we set $\alpha = 0.05$ following CIFAR-10. As shown in Table 2, for this relatively simple model, the results of OTTT and BPTT are similar and BPTT performs slightly better.

Table 2: Performance on Fashion-MNIST.

| Method | Network structure | Time steps | Accuracy |
|---|---|---|---|
| ST-RSBP [17] | 400 (R400) | 400 | 90.00±0.14% (90.13%) |
| IDE [1] | 400 (R400) | 5 | 90.07±0.10% (90.25%) |
| BPTT | 400 (R400) | 5 | 90.58% |
| OTTT$_A$ (ours) | 400 (R400) | 5 | 90.36% |
| OTTT$_O$ (ours) | 400 (R400) | 5 | 90.40% |

## E  Discussion of Limitations and Social Impacts

This work focus on online training of spiking neural networks, and therefore limits the usage of some techniques on network structures such as batch normalization along the temporal dimension. In this work, we adopt the scaled weight standardization as an alternative, which may require additional regularization to fully catch up the best performance of batch normalization as shown in the results of ANNs [4]. It may require exploration of more techniques that is specific for SNNs to improve the performance and meanwhile compatible with more natural properties of SNNs, e.g. the online property.

As for social impacts, since this work focuses only on training methods for spiking neural networks, there is no direct negative social impact. And we believe that the development of successful energy-efficient SNN models could broader its applications and alleviate the huge energy consumption by ANNs. Besides, understanding and improving the training of biologically plausible SNNs may also contribute to the understanding of our brains and bridge the gap between biological neurons and successful deep learning.