# OpenReview forum: "Online Training Through Time for Spiking Neural Networks"
_NeurIPS.cc/2022/Conference — NeurIPS 2022 Accept_

### Official Review · Reviewer_AZJQ · 2022-07-09

**Rating:** 5
**Confidence:** 3
**Soundness:** 1 poor
**Presentation:** 1 poor
**Contribution:** 2 fair

**Summary:**

The paper presents online training through time (OTTT) for SNNs, which is derived from BPTT to enable learning without backward passes through time. The goal compared to other approaches is to improve memory consumption, biological plausibility, or latency. Theoretical analysis shows that the gradients of OTTT are in a direction that is suitable for optimization. Experiments show that, compared to BPTT, OTTT has an advantage in terms of memory consumption, especially as training sequence lengths increase.

**Questions:**

Could the authors explain what are their expectations from the comparisons and analyses that are missing, if they were performed?
Could these actually be peformed?

**Limitations:**

There is no discussion of limitations by the authors.

**Strengths And Weaknesses:**

The paper addresses a topic that has attracted a lot of interest by the community. It also includes significant theoretical analysis and has given effort in explaining the derivations.

On the other hand, the paper:
- does not even mention some recent methods (FPTT, OSTL) [1, 2] that have similar goals, let alone explain the differences. Therefore the extent of the advance in this paper cannot be evaluated.
- does not provide experimental comparisons to such methods as [1-3], but only to BPTT
- if any advantage - theoretical or other - exists compared to these prior methods, the paper does not summarise clearly what element in the new theoretical derivation enables that advantage.
- the work does not aim to achieve something brand new, as successful SNN training has already been possible. Therefore its impact is expected to be limited.
- does not show results on sequential tasks, but only on image classification, even though the learning rule concerns recurrent networks
- does not compare experimentally with non-spiking networks, in terms of accuracy, memory, or computational efficiency.
- could improve its writing, by removing redundant repetitions, providing a sketch/summary of the derivation before the derivation, and by clarifying terms such as forward-in-time

[1] Yin, Bojian, Federico Corradi, and Sander M. Bohte. "Accurate online training of dynamical spiking neural networks through Forward Propagation Through Time." arXiv preprint arXiv:2112.11231 (2021).
[2] Bohnstingl, Thomas, et al. "Online spatio-temporal learning in deep neural networks." IEEE Transactions on Neural Networks and Learning Systems (2022).
[3] Bellec, Guillaume, et al. "A solution to the learning dilemma for recurrent networks of spiking neurons." Nature communications 11.1 (2020): 1-15.

---

> ### Author Response · Authors · 2022-08-02
> **Response to Reviewer AZJQ (part 4/4)**
>
> 7. Suggestions for the writing.
>
> Thank you for your suggestions. We will carefully refine our presentation. The term "forward-in-time" means we only need to do online calculations through time without computing backward through time. This term is similarly used in the literature [16] and we have clarified it in the revision.
>
> 8. Limitation. We have discussed the limitations of the work in Appendix E.
>
> [1] Zheng et al. Going deeper with directly-trained larger spiking neural networks. AAAI, 2021.
>
> [2] Li et al. Differentiable spike: Rethinking gradient-descent for training spiking neural networks. NeurIPS, 2021.
>
> [3] Fang et al. Deep residual learning in spiking neural networks. NeurIPS, 2021.
>
> [4] Deng et al. Temporal Efficient Training of Spiking Neural Network via Gradient Re-weighting. ICLR, 2022.
>
> [5] Wu et al. Spatio-temporal backpropagation for training high-performance spiking neural networks. Frontiers in Neuroscience, 2018.
>
> [6] Zhang and Li. Temporal spike sequence learning via backpropagation for deep spiking neural networks. NeurIPS, 2020.
>
> [7] Fang et al. Incorporating learnable membrane time constant to enhance learning of spiking neural networks. ICCV, 2021.
>
> [8] Xiao et al. Training feedback spiking neural networks by implicit differentiation on the equilibrium state. NeurIPS, 2021.
>
> [9] Amir et al. A low power, fully event-based gesture recognition system. CVPR, 2017.
>
> [10] Shrestha and Orchard. Slayer: Spike layer error reassignment in time. NeurIPS, 2018.
>
> [11] Kaiser et al. Synaptic plasticity dynamics for deep continuous local learning (DECOLLE). Frontiers in Neuroscience, 2020.
>
> [12] Bellec et al. Long short-term memory and learning-to-learn in networks of spiking neurons. NeurIPS, 2018.
>
> [13] Bellec et al. A solution to the learning dilemma for recurrent networks of spiking neurons. Nature Communications, 2020.
>
> [14] Bohnstingl et al. Online spatio-temporal learning in deep neural networks. IEEE Transactions on Neural Networks and Learning Systems, 2022.
>
> [15] Kim et al. Neural architecture search for spiking neural networks. arXiv preprint arXiv:2201.10355.
>
> [16] Neftci et al. Surrogate gradient learning in spiking neural networks: Bringing the power of gradient-based optimization to spiking neural networks. IEEE Signal Processing Magazine, 2019.

---

> ### Author Response · Authors · 2022-08-02
> **Response to Reviewer AZJQ (part 3/4)**
>
> 5. "does not show results on sequential tasks."
>
> First, we would like to point out that our experiment on the neuromorphic dataset DVS-CIFAR10 is **indeed** sequential. The dataset contains sequences of dynamic inputs produced by DVS cameras, which is commonly used to measure the neuromorphic computing of SNNs.
> Second, our experiments follow a large number of previous works on SNNs [1,2,3,4,5,6,7,8] that focus on the most commonly studied static and neuromorphic datasets.
>
> Additionally, we supplement an experiment on another commonly used neuromorphic dataset DVS128-Gesture [9], which contains 11 kinds of hand gestures recorded by a DVS camera. These neuromorphic data are also sequential. The results are below:
>
> | Method | Network structure | Time steps | Accuracy |
> | :----: | :----: | :----: | :----: |
> | SLAYER [10] | 8-layer CNN | 300 | 93.64$\pm$0.49\% |
> | DECOLLE [11] | 3-layer CNN | 1800 | 95.54$\pm$0.16\% |
> | BPTT [7] | 8-layer CNN (PLIF, BN) | 20 | 97.57\% |
> | BPTT [7] | 8-layer CNN (LIF, BN) | 20 | 96.88\% |
> | FPTT (Yin et al., 2021) | 8-layer CNN (LTC-SNN) | 20 | 97.22\% |
> | BPTT | VGG (sWS) | 20 | 96.88\% |
> | OTTT$_A$ (ours) | VGG (sWS) | 20 | 96.88\% |
>
> It shows that our method can achieve the same high performance as BPTT does. The SOTA result [7] incorporates additional techniques to learn membrane time constant and Yin et al. (2021) leverages a more complex neuron model, while we do not dive into such techniques (actually there are only 288 test samples and the 0.69\% accuracy gap stands for 2 samples).
>
> We note that some other works [12,13,14] conduct experiments on other sequential tasks such as speech recognition. However, they show that such tasks require special design for neuron models and architectures to achieve better results [12, 13]. Given the limited time, we are unable to thoroughly dive into them, and it would be important future work. And the recurrence in our work is verified by introducing feedback connections to improve performance on image classification tasks, which is supported by previous works [8, 15].
>
> 6. "does not compare experimentally with non-spiking networks, in terms of accuracy, memory, or computational efficiency."
>
> We supplement the results on CIFAR-10 and CIFAR-100 below. The non-spiking ANN models are based on the ReLU activation instead of spiking neurons.
>
> Results on CIFAR-10 (the last line is ANN):
>
> | Method | Network structure | Params | Time steps | Accuracy |
> | :----: | :----: | :----: | :----: | :----: |
> | ANN-SNN | VGG-16 | 40M | 16 | (92.29\%) |
> | BPTT | ResNet-19 (tdBN) | 14.5M | 6 | (93.16\%) |
> | BPTT | 9-layer CNN (PLIF, BN) | 36M | 8 | (93.50\%) |
> | BPTT | VGG (sWS) | 9.2M | 6 | 92.78$\pm$0.34\% (93.23\%) |
> | OTTT$_A$ (ours) | VGG (sWS) | 9.2M | 6 | 93.52$\pm$0.06\% (93.58\%) |
> | OTTT$_O$ (ours) | VGG (sWS) | 9.2M | 6 | 93.49$\pm$0.17\% (93.73\%) |
> | **ANN** | VGG (sWS) | 9.2M | N.A. | (94.43\%) |
>
> Results on CIFAR-100 (the last line is ANN):
>
> | Method | Network structure | Params | Time steps | Accuracy |
> | :----: | :----: | :----: | :----: | :----: |
> | ANN-SNN | VGG-16 | 40M | 400-600 | (70.55\%) |
> | Hybrid Training | VGG-11 | 36M | 125 | (67.87\%) |
> | DIET-SNN | VGG-16 | 40M | 5 | (69.67\%) |
> | BPTT | VGG (sWS) | 9.3M | 6 | 69.06$\pm$0.07\% (69.15\%) |
> | OTTT$_A$ (ours) | VGG (sWS) | 9.3M | 6 | 71.05$\pm$0.04\% (71.11\%) |
> | OTTT$_O$ (ours) | VGG (sWS) | 9.3M | 6 | 71.05$\pm$0.06\% (71.11\%) |
> | **ANN** | VGG (sWS) | 9.3M | N.A. | (73.19\%) |
>
> Due to the limited time, we are unable to provide ImageNet results. For the neuromorphic dataset DVS-CIFAR10, the equivalent feedforward non-spiking ANNs may not directly handle the dynamic inputs. So we do not consider it. Usually, SNNs with a very small number of time steps do not reach the performance of equivalent ANNs due to the information propagation with discrete spikes rather than floating-point numbers. The results of our model with 6 time steps are acceptable.
>
> Our training memory cost is $O(n)$ (where $n$ is the number of neurons) and is the same as non-spiking ANNs. As for computational efficiency, it is common to compare the energy efficiency between SNNs with spike-based operations and ANNs with floating-point calculations. As has been demonstrated in Section 5.6, with 6 time steps each neuron in our trained model averagely generates 1.1 spikes. Therefore the total synaptic operations of our SNN model would be about the same as the FLOP operations of ANN. Since the cost of synaptic operation is much lower than FLOP operation (this depends on neuromorphic hardware, some can achieve one to two orders of improvement), our SNN model would require much less energy consumption than non-spiking ANNs. Moreover, we can also flexibly reduce the time steps to achieve a trade-off between accuracy and energy consumption, as discussed in Section 5.5 and Appendix D.

---

> ### Author Response · Authors · 2022-08-02
> **Response to Reviewer AZJQ (part 2/4)**
>
> 2. "Does not provide experimental comparisons to such methods as [1-3], but only to BPTT."
>
> All these methods do not scale to large-scale tasks such as ImageNet classification, and for experiments in our work, they hardly have results to compare. Currently, BPTT with SG is the only direct training method that can scale to such large-scale tasks. Therefore, we mainly compare with BPTT and other methods that can achieve high performance on these tasks in the paper. Yin et al. (2021) conduct an experiment on the moderate-scale task DVS-CIFAR10 that is also used in our work. Their result is not better than ours, as listed below:
>
> | Method | Accuracy |
> | :----: | :----: |
> | FPTT (Yin et al., 2021) | 72.3\% |
> | OTTT$_A$ (ours) | 76.27$\pm$0.15\% |
> | OTTT$_O$ (ours) | 76.63$\pm$0.34\% |
>
> They also conduct an experiment on DVS128-Gesture, on which we supplement our results in the following response to question 5. We compare their results in that part.
>
> 3. "If any advantage - theoretical or other - exists compared to these prior methods, the paper does not summarise clearly what element in the new theoretical derivation enables that advantage."
>
> As discussed in the detailed response to question 1, our method is simpler and more efficient to implement, provides a more solid theoretical grounding, and performs better even on large-scale datasets. Our new theoretical derivation is unique and enables a clearer explanation of the optimization. In detail, we do not try to seek the exact equivalence to gradients calculation by BPTT with SG which is unclear for optimization considering the non-differentiability, but we connect OTTT with another branch of SNN training methods, i.e. methods based on spike representation which is theoretically more clear for optimization, for theoretical analysis and guarantee.
>
> 4. "The work does not aim to achieve something brand new, as successful SNN training has already been possible. Therefore its impact is expected to be limited."
>
> We respectfully disagree with this statement. The training method for SNNs is still an important open problem, especially if we want to consider more properties that are suitable for on-chip learning on neuromorphic hardware. As for existing training methods, while they may be successful regarding the performance (e.g. ANN-SNN or BPTT with SG, and the direct training methods only scale to large-scale datasets very recently [1,2,3,4] and may still have a large improvement space), they all have important limitations as we have introduced in the paper, e.g. BPTT with SG suffers from large training memory costs and lack of theoretical clarity for optimization, and they are inconsistent with the online property of rules on hardware. As for existing online training methods, as discussed in the response to the first question, from the theoretical perspective, they lack solid theoretical grounding for optimizing non-differentiable SNNs, and from the experimental perspective, none of them scale to large-scale tasks with large networks structures as BPTT with SG do. And some of these methods require more memory costs and could be more complex to be implemented than our method, e.g. on neuromorphic hardware. So it is still an important problem to study proper SNN training methods.
>
> Our method and analysis should make important contributions from both theoretical and practical perspectives. As discussed in the detailed response to question 1, our method provides a more solid theoretical grounding for optimizing non-differentiable SNNs, and our online method can scale to large-scale tasks with more efficient computation and less costs, which could also pave a path for online on-chip training of SNNs. Note that Reviewer Mcuq also pointed out in "Strengths" that "This article can form a substantial step in that direction (for online learning on neuromorphic hardware)."

---

> ### Author Response · Authors · 2022-08-02
> **Response to Reviewer AZJQ (part 1/4)**
>
> Thank you for your comments. We try our best to address your concerns as follows.
>
> 1. About the related work Yin et al. (2021) and Bohnstingl et al. (2022), and the difference between our work and theirs.
>
> We provide a very detailed discussion below, and we have added the citation and discussion in the revised paper.
>
> Yin et al. (2021) is a recent work that directly leverages the RNN training method named forward propagation through time (FPTT) [1] to train spiking neural networks with the help of surrogate gradients (SG). Moreover, they propose a new liquid spiking neuron whose time constant depends on the input and previous membrane potentials, and show that FPTT should be combined with this neuron for good results.
>
> Bohnstingl et al. (2022) is a recent work that proposes an online learning method OSTL for recurrent and spiking neural networks, and for non-differentiable SNNs, they also use surrogate gradients.
>
> Our work is different from them in three main aspects.
>
> (1) **Our training method is simpler and more efficient than them.**
>
> For FPTT, the original FPTT [1] trains recurrent neural networks by dynamically regularizing weights. It calculates gradients at each time step based on the current state, and regularizes the update of weights by a penalty loss which is based on the running average of previous weights and the previous gradient. Yin et al. (2021) directly apply this method to SNNs and require heavy computation to regularize the update of parameters. As a comparison, we calculate gradients based on the tracked pre-synaptic activities and only need to update parameters according to simple rules, which is computationally efficient and could be easier to be implemented, e.g. on neuromorphic hardware.
>
> For OSTL, they seek the exact equivalence with BPTT with SG, so their tracked eligibility traces have a much larger memory overhead than our tracked pre-synaptic activities. Particularly, the memory complexity of their method is $O(n^2)$, while ours is only $O(n)$, where $n$ is the number of neurons in a layer (the complexity for BPTT is $O(Tn)$, where $T$ is the time steps). This is because they consider the derivative of the reset operation so that they have to maintain a large tensor for eligibility traces, while we do not consider this so the derivation can be simplified to Eq. (4) in the paper, and therefore we only need to track pre-synaptic activities for each neuron. So our method is much simpler and requires fewer costs (the training memory costs advantage has been demonstrated in Fig. 2, which shows that we reduce the $O(Tn)$ complexity of BPTT to $O(n)$, rather than increasing to $O(n^2)$), and our tracked trace could be easier to be implemented on neuromorphic hardware.
>
> (2) **Our method has a more solid theoretical grounding for optimization.**
>
> The major obstacle of training SNNs is that the spiking operation is discrete and non-differentiable. Therefore, directly applying RNN training methods to SNNs is problematic as the derivative of the Heaviside step function is 0 almost everywhere. Previous works that apply learning methods of RNNs to SNNs (including Yin et al. (2021)) or seek the exact equivalence with these learning methods (including Bohnstingl et al. (2022)) use "surrogate gradients" (SG) to handle this problem, which substitutes the derivative of the step function with continuous approximations. However, gradient descent with such a method in the context of RNN-like training typically lacks theoretical clarity for optimization, since it is not the true gradient of the actual function and the descent direction is not guaranteed.
>
> Unlike these works, we provide a more solid theoretical grounding from a new perspective. We do not try to seek the exact equivalence to gradients calculation by BPTT (or similar methods for RNN) with SG. Instead, we connect OTTT with another branch of SNN training methods, i.e. methods based on spike representation which is better for theoretical analysis of optimization. This branch of methods builds the connection between spike representation (e.g. the (weighted) firing rate or spiking time) of neurons in an ANN-like closed form that is sub-differentiable. So gradients can be calculated through the spike representation and are well defined. We prove that gradients of OTTT can provide a similar descent direction as these gradients based on spike representation and therefore provide a theoretical grounding for optimization in the context of the optimization problem formulated by spike representation. This also provides a connection between the two mainstream SNN training methods, i.e. BPTT with SG and spike representation-based methods.
>
> (3) **Our experiments can scale to large-scale tasks** including ImageNet classification and are based on the common LIF neuron, while Yin et al. (2021) and Bohnstingl et al. (2022) do not consider the large-scale problem, and Yin et al. (2021) require a more complex neuron model.

---

> ### Author Response · Authors · 2022-08-07
> **To Reviewer AZJQ: Could you please check out our response and re-evaluate our paper?**
>
> Dear Reviewer AZJQ,
>
> Thank you again for your review. We have tried our best to address your concerns in the response / the updated paper with very detailed discussions and additional results. Since the author-reviewer discussion deadline is approaching, could you please take a look at our response and re-evaluate our paper? We are willing to answer questions if you have other concerns. Thank you for your consideration and we are looking forward to hearing from you.
>
> Sincerely,
>
> Authors of Paper 1731

---

> > ### Comment · Reviewer_AZJQ · 2022-08-07
> > **Update**
> >
> > Thank you for the responses, and apologies for the delay in replying.
> > The earlier weaknesses were important, but the authors have addressed them rather well.
> > I am updating the rating.
> > I would recommend that the authors reflect in the main text, not only the appendix, some of the important new clarifications and results..

---

### Official Review · Reviewer_Mcuq · 2022-07-11

**Rating:** 7
**Confidence:** 3
**Soundness:** 3 good
**Presentation:** 3 good
**Contribution:** 3 good

**Summary:**

The authors introduce a new approach to training spiking neural networks, online training through time (OTTT). The approach only requires constant memory since it does not need to backpropagate through time. Instead, tracks presynaptic activities and leverages instantaneous loss and gradients. The proposed approach provides good results on typical benchmarks such as CIFAR-10 and 100.



**Questions:**

1. How does the authors' work relate to other approaches that implement forward propagation through time?

2.  What are the effects of the assumptions on the proof? For instance, in line 167 the reset moment is ignored. Line 147-150: Does the equilibrium condition depend on constant inputs? And why can only equilbria happen and not limit cycles?

3. The "instantaneous" loss is defined as a sum over time (lie 184). How is that instantaneous and the definition higher up (line 182) not? They both sum over time, although 1/T is inside the loss function higher up.

4. Intuitively, what is remembered in the trace that allows to bypass remembering all activations over time?

5. What are the gradients of the "spike representation"? Are these the gradients from BPTT?

6. What is meant with "time steps" in Fig 2? The time steps BPTT goes back in time? How does this apply to OTTT?


**Strengths And Weaknesses:**

Strengths:
* How to best train SNNs is still an open problem, especially if one aims for online learning on neuromorphic hardware. This article can form a substantial step in that direction.
* If I understand the mathematical proof well, the authors show that their method's gradient points in the same direction as the BPTT one, without needing to track many variables in memory.
* The results of the approach seem good.

Weaknesses:
* The work currently does not discuss other SNN  approaches that also do not backpropagate gradients through time, such as: Accurate online training of dynamical spiking neural networks through Forward Propagation Through Time, B Yin, F Corradi, SM Bohte - arXiv preprint arXiv:2112.11231, 2021https://arxiv.org/pdf/2112.11231.pdf
* Some of the main intuitions behind not needing to backpropagate through time remain unclear (see questions).
*  The tasks to which the approach is applied are quite "static", and do not immediately seem to need much memory. Moreover, the approach seems to rely on neurons moving to an equilibrium state, which would be less valid for real-world, time-varying data.

---

> ### Author Response · Authors · 2022-08-02
> **Response to Reviewer Mcuq (part 3/3)**
>
> 6. "What are the gradients of the "spike representation"?"
>
> The gradients of the spike representation are gradients calculated through the closed-form transformation between the spike representation of neurons in different layers, which are not the gradients from BPTT. In this work, we consider the weighted firing rate as spike representation: $\mathbf{a}[t]=\frac{\sum_{\tau=0}^t \lambda^{t-\tau}\mathbf{s}[\tau]}{\sum_{\tau=0}^t \lambda^{t-\tau}}$, and the closed-form transformation is $\mathbf{a}^{l+1}[T] \approx \sigma\left(\frac{1}{V_{t h}}\left(\mathbf{W}^{l} \mathbf{a}^{l}[T]+\mathbf{b}^{l+1}\right)\right)$. The gradients of spike representation are calculated as $\frac{\partial L}{\partial \mathbf{W}^l}=\frac{\partial L}{\partial \mathbf{a}^N[T]}\prod_{i=N-1}^{l+1}\frac{\partial \mathbf{a}^{i+1}[T]}{\partial \mathbf{a}^i[T]}\frac{\partial \mathbf{a}^{l+1}[T]}{\partial \mathbf{W}^l}$ (refer to the "Spike Representation" paragraph in Section 3.2). As explained in the paper and the second point in the above response to question 1, direct gradients from BPTT for non-differentiable SNNs are problematic, and BPTT with surrogate gradients is not theoretically clear for optimization. We build the connection between gradients of OTTT and gradients based on spike representation (which is more theoretically clear) to prove the descent guarantee for the optimization problem.
>
> 7. "What is meant with "time steps" in Fig 2?"
>
> "Time Steps" are the discrete time steps to simulate SNNs. The simulation of forward SNN dynamics is discretized and unfolded over time, and it applies to SNN models regardless of training methods.
> Fig 2 is to show the memory cost comparison between BPTT and OTTT under different settings of the simulation time steps of SNNs.
>
> [1] Kag and Saligrama. Training recurrent neural networks via forward propagation through time. ICML, 2021.
>
> [2] Xiao et al. Training feedback spiking neural networks by implicit differentiation on the equilibrium state. NeurIPS, 2021.
>
> [3] Amir et al. A low power, fully event-based gesture recognition system. CVPR, 2017.
>
> [4] Shrestha and Orchard. Slayer: Spike layer error reassignment in time. NeurIPS, 2018.
>
> [5] Kaiser et al. Synaptic plasticity dynamics for deep continuous local learning (DECOLLE). Frontiers in Neuroscience, 2020.
>
> [6] Fang et al. Incorporating learnable membrane time constant to enhance learning of spiking neural networks. ICCV, 2021.

---

> > ### Comment · Reviewer_Mcuq · 2022-08-07
> > **Response**
> >
> > Thank you for the explanations.
> >
> > Re 7: So, with "time steps" you mean the total run-time of the network, where it is reset at time step 0 and you present (constant?) inputs for the mentioned number of time steps? And you assume that BPTT is unrolled fully over time? In my opinion this requires some clarification in the text.
> >
> > I remain at "accept" as my evaluation, I think the current paper presents an interesting avenue for SNN learning, which is in my eyes not yet a solved issue.

---

> > > ### Author Response · Authors · 2022-08-07
> > > **Response to Reviewer Mcuq**
> > >
> > > Thank you for the valuable suggestion and we will clarify this in the following revision. Yes, for each input sample, the network is reset at time step 0, and at each discrete time step $t$ the input at time step $t$ is passed to the network, with total $T$ time steps. For static images, the input at all time steps is the real-valued image pixel value, which is similar to previous work and can be regarded as the input current. For dynamic inputs, the input at each time step is different. This is the setting in many previous works and their realization of BPTT is to unroll over these discrete time steps. We follow this setting and we will add the clarification.

---

> ### Author Response · Authors · 2022-08-02
> **Response to Reviewer Mcuq (part 2/3)**
>
> 2. "The tasks are quite "static", and do not immediately seem to need much memory. The approach would be less valid for real-world, time-varying data."
>
> In our experiments, the inputs of image classification tasks including CIFAR-10, CIFAR-100, and ImageNet are static, while the inputs of the neuromorphic dataset DVS-CIFAR10 are dynamic, which is converted from CIFAR-10 by DVS cameras. Our theorems apply for convergent inputs (i.e., the weighted average of the input sequence converges), so it can work well on time-varying but convergent data such as DVS-CIFAR10.
>
> Furthermore, we supplement an additional experiment on DVS128-Gesture [3], which contains 11 kinds of hand gestures recorded by a DVS camera and is more real-world and time-varying. The results are below:
>
> | Method | Network structure | Time steps | Accuracy |
> | :----: | :----: | :----: | :----: |
> | SLAYER [4] | 8-layer CNN | 300 | 93.64$\pm$0.49\% |
> | DECOLLE [5] | 3-layer CNN | 1800 | 95.54$\pm$0.16\% |
> | BPTT [6] | 8-layer CNN (PLIF, BN) | 20 | 97.57\% |
> | BPTT [6] | 8-layer CNN (LIF, BN) | 20 | 96.88\% |
> | BPTT | VGG (sWS) | 20 | 96.88\% |
> | OTTT$_A$ (ours) | VGG (sWS) | 20 | 96.88\% |
>
> It shows that our method is also applicable to such time-varying data and achieves the same high performance as BPTT. The SOTA result [6] incorporates additional techniques to learn membrane time constant, while we do not dive into such techniques (actually there are only 288 test samples and the 0.69\% accuracy gap stands for 2 samples). While our theorems mainly consider convergent inputs, it would be interesting future work to further consider the theoretical grounding for time-varying non-convergent inputs.
>
> Besides, the memory costs in our comparison are not for inputs but for intermediate variables of unfolded SNNs during training. Even for static inputs, BPTT for SNNs should maintain the computational graph unfolded along the simulation time, while our OTTT does not, and this also holds for dynamic inputs.
>
> 3. About "the effects of the assumptions on the proof".
>
> (1) *"The reset moment is ignored"*. This is a part of our method rather than an assumption. The reset operation is ignored so that we can track pre-synaptic activities of each neuron to decouple the temporal dependency. This further enables our gradients to be aligned with gradients by spike representation, which supports the proof of the descent direction. Note that training non-differentiable SNNs by BPTT with SG is theoretically unclear, so we do not seek the exact equivalence with the form of BPTT but build the connection with gradients based on spike representation and prove the descent guarantee for the optimization problem (as explained in the paper and the second point in the above response to question 1).
>
> (2) *"Does the equilibrium condition depend on constant inputs? And why can only equilibria happen and not limit cycles?"* The equilibrium condition depends on constant or convergent inputs, i.e. the weighted average inputs $\mathbf{\overline{x}}[t]=\frac{\sum_{\tau=0}^t \lambda^{t-\tau}\mathbf{x}[\tau]}{\sum_{\tau=0}^t \lambda^{t-\tau}}$ converge through time $\mathbf{\overline{x}}[t]\rightarrow \mathbf{x^*}$ (line 142 in the paper). The equilibrium (but not limit cycle) would happen with contractive recurrent connections. This is proven in [2] according to the contractive mapping theorem, and we also consider the contractive recurrent connections.
>
> 4. About the instantaneous loss.
>
> We would like to clarify that the instantaneous loss at time $t$ is $L[t]=\frac{1}{T}\mathcal{L}\left(\mathbf{s}^N[t], \mathbf{y}\right)$, and considering all time steps the total loss is $L\coloneqq\sum_{t=1}^TL[t]$. So the instantaneous loss can be computed independently at each time step, while the loss based on firing rate depends on all time steps and does not support online gradients. We have clarified this in the revision.
>
> 5. "Intuitively, what is remembered in the trace that allows to bypass remembering all activations over time?"
>
> Intuitively, the tracked pre-synaptic activities $\hat{\mathbf{a}}^l[t] = \sum_{\tau \leq t}\lambda^{t-\tau}\mathbf{s}^l[\tau]$ maintain the previous spikes by different coefficients that are related to the time constant of the LIF neuron, then propagation through the trace could deal with previous spikes.

---

> > ### Comment · Reviewer_Mcuq · 2022-08-07
> > **Response**
> >
> > Thank you for the clarifications.
> >
> > Re 2: To clarify my point: Although the inputs for tasks come from DVS cameras, the outputs are classifications, which are static over time. Patterns relevant for classification can be discernable on quite short time scales, hence not requiring any long-term memory. I do realize that time and space for the paper are limited, and application to e.g. reinforcement learning tasks can be left to future work.
> >
> > Re 4: if I understand correctly now, it depends on a spike occurring at time t or not. It then seems to me that this method is mostly applicable to classification tasks, where typically one neuron needs to be 1 and the rest 0. Can the method also be applied to regression?

---

> > > ### Author Response · Authors · 2022-08-07
> > > **Response to Reviewer Mcuq**
> > >
> > > Thank you for the valuable comments.
> > >
> > > We agree that the outputs of the classification task are static over time, and it would be interesting future work to consider applications such as reinforcement learning. Many SNN models do not explicitly model long-term memory and additional efforts on the model (e.g. specifically designed architecture) are required for these tasks. We will consider extending the training method to these scenarios in future work.
> > >
> > > As for the regression task, our method is also applicable. For SNNs, regression is typically done with the firing rate as well. So the output and the loss are similar to that of the classification task, i.e. the loss is $L=\mathcal{L}(\frac{1}{T}\sum_{t=1}^T\mathbf{s}^N[t], \mathbf{y})$, where $\mathbf{y}$ would be the regression target, and the total loss of our instantons loss is still an upper bound of this loss. Also, note that the output of the model does not need to be restricted to 0 or 1. In practice, we assume that the neurons of the last output layer will not spike or reset and do classification based on the accumulated membrane potential, which is similar to previous works (see Appendix C.2.2). So the output is $\mathbf{u}^N[t]$, which is not restricted to binary output, and the loss is calculated between $\mathbf{u}^N[t]$ and $y$.

---

> ### Author Response · Authors · 2022-08-02
> **Response to Reviewer Mcuq (part 1/3)**
>
> Thank you very much for appreciating our work. We respond to your valuable comments as follows.
>
> 1. About the related work Yin et al. (2021), and the difference between our work and theirs.
>
> Thanks for the reference. Yin et al. (2021) is a recent work that directly leverages the RNN training method named forward propagation through time (FPTT) [1] to train spiking neural networks with the help of surrogate gradients. Moreover, they propose a new liquid spiking neuron whose time constant depends on the input and previous membrane potentials and show that FPTT should be combined with this neuron for good results.
>
> Our work is different from theirs in three main aspects.
>
> (1) **Our training method is simpler and more efficient than FPTT.**
>
> The original FPTT [1] trains recurrent neural networks by dynamically regularizing weights. It calculates gradients at each time step based on the current state and regularizes the update of weights by a penalty loss based on the running average of previous weights and the previous gradient. Yin et al. (2021) directly apply this method to SNNs and require heavy computation to regularize the update of parameters. As a comparison, we calculate gradients based on the tracked pre-synaptic activities and only need to update parameters according to simple rules, which is computationally efficient and could be easier to be implemented, e.g. on neuromorphic hardware.
>
> (2) **Our method has a more solid theoretical grounding for optimization.**
>
> The major obstacle to training SNNs is that the spiking operation is discrete and non-differentiable. Therefore, directly applying RNN training methods to SNNs is problematic as the derivative of the Heaviside step function is 0 almost everywhere. Previous works that apply learning methods of RNNs to SNNs (including Yin et al. (2021)) use "surrogate gradients" (SG) to handle this problem, which substitutes the derivative of the step function with continuous approximations. However, gradient descent with such a method in the context of RNN-like training typically lacks theoretical clarity for optimization, since it is not the true gradient of the actual function, and the descent direction is not guaranteed.
>
> Unlike these works, we provide a more solid theoretical grounding from a new perspective. We do not try to seek the exact equivalence to gradients calculation by BPTT (or similar methods for RNN) with SG. Instead,  we connect OTTT with another branch of SNN training methods, i.e., methods based on spike representation which is better for theoretical analysis of optimization. This branch of methods builds the connection between spike representation (e.g., the (weighted) firing rate or spiking time) of neurons in an ANN-like closed form that is sub-differentiable. So gradients can be calculated through the spike representation and are well defined. We prove that gradients of OTTT can provide a similar descent direction as these gradients based on spike representation and therefore provide a theoretical grounding for optimization in the context of the optimization problem formulated by spike representation. This also provides a connection between the two mainstream SNN training methods, i.e., BPTT with SG and spike representation-based methods.
>
> (3) **Our experiments can scale to large-scale tasks** including ImageNet classification and are based on the common LIF neuron, while Yin et al. (2021) do not consider the large-scale problem and require a more complex neuron model.
>
> We have added the citation and discussion in the revised paper.

---

### Official Review · Reviewer_bMnW · 2022-07-12

**Rating:** 7
**Confidence:** 4
**Soundness:** 4 excellent
**Presentation:** 3 good
**Contribution:** 3 good

**Summary:**

In this paper, the authors propose an online learning algorithm (OTTT) that's applicable to spiking neural networks (SNNs). OTTT uses a combination of eligibility traces (trace of past activity of the neuron) and use of instantaneous loss values to achieve an online algorithm. The authors show its connection to spike representation based methods as well as three-factor learning rule and demonstrate empirically that this method works better than existing methods for training feedforward spiking neural networks.


**Questions:**

## Questions

* Does $s[\tau]$ in line 141 refer to the output spikes of each layer?
* Can OTTT also be applied to non-spiking neural network architectures?

## Suggestions

* In the abstract:
    * l.5: it's not clear why BPTT with SG leads to extremely low latency. Rephrase?
    * l.6: theoretical unclarity -> lack of theoretical clarity.
    * l.11: what does it mean for online learning to be a learning rule? Rephrase?
* l.32: The supervised training of SNNs is challenging not because of its "complex neuron model" but because of non-differentiability in the neuron model
* Sec 2.  I would suggest giving brief descriptions of individual references rather than grouping large numbers of references together with vague sentences, which doesn't convey much useful information about related work.
* Related work that uses eligibility traces should be cited: e.g. (Murray 2019; Bellec et al. 2020)
* l.142: A few details of how this can be proved would be useful.
* Table 1: It would be helpful to see comparison with equivalent non-spiking architecture trained with BPTT as an (upper) baseline to put the performance of OTTT in perspective.

Bellec, G., Scherr, F., Subramoney, A., Hajek, E., Salaj, D., Legenstein, R., Maass, W., 2020. A solution to the learning dilemma for recurrent networks of spiking neurons. Nature Communications 11, 3625.

Murray, J.M., 2019. Local online learning in recurrent networks with random feedback. eLife 8, e43299.

**Limitations:**

The authors do not discuss the limitations of the work.

**Strengths And Weaknesses:**

## Strengths

The paper derives an novel online learning rule, primarily applied to feed-forward SNNs. The connection to spike representation derived in the paper is very interesting, and connects these seemingly disparate methods.

The empirical results are impressive, and strongly support the utility of OTTT. The fact that OTTT works with batch size of 1 is particularly impressive, and makes it of strong practical interest for neuromorphic hardware.

The paper is also well written and the exposition is clear and easy to understand.

## Weaknesses

The results of OTTT with feedback connections and with smaller batch sizes needs more baselines, esp. compared to BPTT to understand how it stands in comparison to conventional methods. i.e. to understand if the advantage there because of OTTT or other reasons?

It would also have been interesting to see results with fully recurrent architectures.

Minor: The connection with spike representation methods is certainly interesting, but might be a bit over-emphasized and could be confined to a single section so that it doesn't interrupt the flow of OTTT. Specifically, the Spike Representation heading in Sec. 3.2 threw me off a bit, since I was looking for a connection with Sec. 4.1.

---

> ### Author Response · Authors · 2022-08-02
> **Response to Reviewer bMnW (part 2/2)**
>
> 7. "Comparison with equivalent non-spiking architecture trained with BPTT as an (upper) baseline".
>
> Thank you for the suggestion. Models in Table 1 are feed-forward networks and therefore the equivalent non-spiking architecture will not be unfolded through time. So they will just be trained by BP. We supplement the results on CIFAR-10 and CIFAR-100 below. The ANN models are based on the ReLU activation instead of spiking neurons.
>
> Results on CIFAR-10 (the last line is ANN):
>
> | Method | Network structure | Params | Time steps | Accuracy |
> | :----: | :----: | :----: | :----: | :----: |
> | ANN-SNN | VGG-16 | 40M | 16 | (92.29\%) |
> | BPTT | ResNet-19 (tdBN) | 14.5M | 6 | (93.16\%) |
> | BPTT | 9-layer CNN (PLIF, BN) | 36M | 8 | (93.50\%) |
> | BPTT | VGG (sWS) | 9.2M | 6 | 92.78$\pm$0.34\% (93.23\%) |
> | OTTT$_A$ (ours) | VGG (sWS) | 9.2M | 6 | 93.52$\pm$0.06\% (93.58\%) |
> | OTTT$_O$ (ours) | VGG (sWS) | 9.2M | 6 | 93.49$\pm$0.17\% (93.73\%) |
> | **ANN** | VGG (sWS) | 9.2M | N.A. | (94.43\%) |
>
> Results on CIFAR-100 (the last line is ANN):
>
> | Method | Network structure | Params | Time steps | Accuracy |
> | :----: | :----: | :----: | :----: | :----: |
> | ANN-SNN | VGG-16 | 40M | 400-600 | (70.55\%) |
> | Hybrid Training | VGG-11 | 36M | 125 | (67.87\%) |
> | DIET-SNN | VGG-16 | 40M | 5 | (69.67\%) |
> | BPTT | VGG (sWS) | 9.3M | 6 | 69.06$\pm$0.07\% (69.15\%) |
> | OTTT$_A$ (ours) | VGG (sWS) | 9.3M | 6 | 71.05$\pm$0.04\% (71.11\%) |
> | OTTT$_O$ (ours) | VGG (sWS) | 9.3M | 6 | 71.05$\pm$0.06\% (71.11\%) |
> | **ANN** | VGG (sWS) | 9.3M | N.A. | (73.19\%) |
>
> Due to limited time, we are unable to provide ImageNet results. For the neuromorphic dataset DVS-CIFAR10, the equivalent feedforward non-spiking ANNs may not directly handle the dynamic inputs. So we do not consider them. Usually, SNNs with a very small number of time steps do not reach the performance of equivalent ANNs due to the information propagation with discrete spikes rather than floating-point numbers. The results of our model with 6 time steps are acceptable.
>
> 8. Limitations. We have discussed the limitations of the work in Appendix E.
>
> [1] Kim et al. Neural architecture search for spiking neural networks. arXiv preprint arXiv:2201.10355.
>
> [2] Zhang and Li. Spike-train level backpropagation for training deep recurrent spiking neural networks. NeurIPS, 2019.
>
> [3] Xiao et al. Training feedback spiking neural networks by implicit differentiation on the equilibrium state. NeurIPS, 2021.

---

> ### Author Response · Authors · 2022-08-02
> **Response to Reviewer bMnW (part 1/2)**
>
> Thank you very much for appreciating our work. We respond to your valuable comments as follows.
>
> 1. "The results of OTTT with feedback connections and with smaller batch sizes need more baselines."
>
> Thank you for the suggestion. We follow your advice and conduct additional experiments to compare our method with the BPTT baseline. The results of feedback connections are below:
>
> | Network structure | Method | Accuracy |
> | :----: | :----: | :----: |
> | VGG | OTTT$_O$ | 71.05$\pm$0.06\% (71.11\%) |
> | VGG-F | OTTT$_O$ | 72.63$\pm$0.23\% (72.94\%) |
> | VGG | BPTT | 69.06$\pm$0.07\% (69.15\%) |
> | VGG-F | BPTT |  (69.49\%) |
>
> First, it can be seen from the table above that using feedback connections improves performance when trained by BPTT with surrogate gradients. Indeed, this is a known fact demonstrated in previous works [1]. Second, we can see that in different settings, compared with BPTT, OTTT consistently achieves higher performance, and the improvement of OTTT from feedback connections is more significant than that of BPTT.
>
> The results of training with batch size 1 are below:
>
> | Method | Batch Size | Accuracy |
> | :----: | :----: | :----: |
> | OTTT$_A$ / OTTT$_O$ | 128 | 88.20\% / 88.62\% |
> | OTTT$_A$ / OTTT$_O$ | 1 | 88.07\% / 88.50\% |
> | BPTT | 1 | 87.51\% |
>
> It shows that a good model can be obtained by BPTT/OTTT with batch size 1, and OTTT performs better. This is because we do not use batch normalization (as explained in Section 4.4), so the model is less sensitive to the batch size.
> Combined with the online-in-time property of OTTT, which would correspond to the temporally local property of biological learning rules and rules on neuromorphic hardware (BPTT does not have this property), such training scenario could pave a path for online on-chip learning.
>
> 2. "It would also have been interesting to see results with fully recurrent architectures."
>
> Thanks for the suggestion again. We conduct an experiment to use a recurrent spiking neural network on the Fashion-MNIST classification task. The input is flattened as a vector with 784 dimensions and is connected to 400 spiking neurons with recurrent connections. The outputs of neurons are then connected to a readout layer for classification. We compare BPTT, OTTT$_A$, and OTTT$_O$ with the results in [2] and [3]. The results are below:
>
> | Method | Time steps | Accuracy |
> | :----: | :----: | :----: |
> | ST-RSBP [2] | 400 | 90.00$\pm$0.14\% (90.13\%) |
> | IDE [3] | 5 | 90.07$\pm$0.10\% (90.25\%) |
> | BPTT | 5 | (90.58\%) |
> | OTTT$_A$ (ours) | 5 | (90.36\%) |
> | OTTT$_O$ (ours) | 5 | (90.40\%) |
>
> From the table, we can see that for this relatively simple model, the results of OTTT and BPTT are very similar and BPTT performs slightly better.
>
> 3. "Does $s[\tau]$ in line 141 refer to the output spikes of each layer?"
>
> Yes, the notation here considers a group of neurons that can refer to each layer, and $s[\tau]$ represents the output spikes of the neurons at time $\tau$.
>
> 4. "Can OTTT also be applied to non-spiking neural network architectures?"
>
> Currently, OTTT is designed for SNNs. The derivation is based on the spiking neuron dynamics and spike signals, and it is to handle the problem of the optimization for non-differentiable SNNs with theoretical guarantee.
>
> 5. About suggestions for writing and organization.
>
> Thank you very much for your valuable suggestions. We have carefully considered them and revised the paper. In this revision, we re-organize the order of Section 3.2 and modify some descriptions.
> Responses to several questions are below:
>
> (1) "It's not clear why BPTT with SG leads to extremely low latency".
>
> Previous works empirically show that training models by BPTT with SG can achieve high performance with a very small number of time steps compared with other methods. This is why we call it "with extremely low latency". We have made the description more precise in this revision.
>
> (2) "What does it mean for online learning to be a learning rule".
>
> We mean that the learning rule is temporally local and has the online property. We modify the description in the revision as "... inconsistent with the online property of biological learning rules and rules on neuromorphic hardware".
>
> 6. "Related work that uses eligibility traces should be cited".
>
> Thank you for pointing it out. We have already cited Bellec et al. (2020) in the originally submitted paper (lines 109 \& 196 in the paper). Murray (2019) uses eligibility traces to train non-spiking recurrent networks. We have added the references in the revision.

---

### Author Response · Authors · 2022-08-02
**A summary of paper updates**

We thank all reviewers for their valuable comments and suggestions. We have uploaded an updated version of our paper based on the reviews. Revisions are marked as blue in the text. The updates are summarized as follows:

1. We add the citation and discussion of the recent related work mentioned by reviewers in Section 2.

2. Following the suggestions from reviewers, we supplement several experiments (provided in the response) in Appendices D.2, D.3, and D.4. Due to the limited space, the supplemented results are currently in Appendix.

3. Following the suggestions from Reviewer bMnW, we re-organize the order of Section 3.2 and modify several descriptions in the Abstract and Introduction.

4. In response to Reviewer Mcuq, we clarify the description of the instantaneous loss in Section 4.1.

5. Following the suggestions from Reviewer AZJQ, we clarify the term "forward-in-time" in Section 1.

We will continually revise the paper based on the suggestions. Thanks for the valuable comments again.

---

### Meta-Review · Area_Chair_nt7Z · 2022-08-24

**Recommendation:** Accept
**Confidence:** Certain

**Metareview:**

The authors propose an online training algorithm (OTTT) for spiking neural networks (SNNs) using eligibility traces and instantaneous loss values. They show empirically that this method performs better than previous ones in feed-forward spiking neural networks.

All reviewers agree that the empirical results are impressive and that the method is interesting for neuromorphic hardware. The authors also provide a mathematical analysis of the learning method.

Weaknesses:
- Networks are mostly applied to static tasks, while more temporal tasks are potentially more interesting for SNNs
- Comparison to previously proposed methods is missing.

In general, a very interesting and strong paper. I propose acceptance.

**Award:**

No

---

### Decision · Program_Chairs · 2022-09-14

Accept